# PyCO2SYS v1.8: marine carbonate system calculations in Python

Matthew P. Humphreys[1], Ernie R. Lewis[2], Jonathan D. Sharp[3,4], and Denis Pierrot[5]

[1]NIOZ Royal Netherlands Institute for Sea Research, Department of Ocean Systems (OCS), Texel, the Netherlands
[2]Environmental and Climate Sciences Department, Brookhaven National Laboratory, Upton, NY, USA
[3]Cooperative Institute for Climate, Ocean, and Ecosystem Studies, University of Washington, Seattle, WA, USA
[4]Pacific Marine Environmental Laboratory, National Oceanic and Atmospheric Administration, Seattle, WA, USA
[5]Atlantic Oceanographic and Meteorological Laboratory, National Oceanic and Atmospheric Administration, Miami, FL, USA

**Correspondence:** matthew.humphreys@nioz.nl

**Abstract.** Oceanic dissolved inorganic carbon ($T_\mathrm{C}$) is the largest pool of carbon that interacts substantially with the atmosphere on human timescales. Oceanic $T_\mathrm{C}$ is increasing through uptake of anthropogenic carbon dioxide ($CO_2$), and seawater pH is decreasing as a consequence. Both the exchange of $CO_2$ between ocean and atmosphere and the pH response are governed by a set of parameters that interact through chemical equilibria, collectively known as the marine carbonate system. To investigate these processes, at least two of the marine carbonate system's parameters are typically measured — most commonly, two from $T_\mathrm{C}$, total alkalinity ($A_\mathrm{T}$), pH, and seawater $CO_2$ fugacity ($f_{CO_2}$; or its partial pressure, $p_{CO_2}$, or its dry-air mole fraction, $x_{CO_2}$) — from which the remaining parameters can be calculated and the equilibrium state of seawater solved. Several software tools exist to carry out these calculations, but no fully functional and rigorously validated tool written in Python, a popular scientific programming language, was previously available. Here, we present PyCO2SYS, a Python package intended to fill this capability gap. We describe the elements of PyCO2SYS that have been inherited from the existing CO2SYS family of software and explain subsequent adjustments and improvements. For example, PyCO2SYS uses automatic differentiation to solve the marine carbonate system and calculate chemical buffer factors, ensuring that the effect of every modelled solute and reaction is accurately included in all its results. We validate PyCO2SYS with internal consistency tests and comparisons against other software, showing that PyCO2SYS produces results that are either virtually identical or different for known reasons, with the differences negligible for all practical purposes. We discuss insights that guided the development of PyCO2SYS, for example that the marine carbonate system cannot be unambiguously solved from certain pairs of parameters. Finally, we consider potential future developments to PyCO2SYS and discuss the outlook for this and other software for solving the marine carbonate system. The code for PyCO2SYS is distributed via GitHub (https://github.com/mvdh7/PyCO2SYS) under the GNU General Public License v3, archived on Zenodo (Humphreys et al., 2021), and documented online (https://PyCO2SYS.readthedocs.io).

## 1 Introduction

The ocean absorbs about a quarter of the anthropogenic carbon dioxide ($CO_2$) currently emitted each year (Friedlingstein et al., 2020). This absorption is a double-edged sword. Removing $CO_2$ from the atmosphere reduces the impact of these emissions on Earth's climate. However, $CO_2$ uptake causes seawater pH and calcium carbonate mineral saturation states ($\Omega$) to decline

through a process termed ocean acidification, which has adverse effects on some marine species and ecosystems (Doney et al., 2009).

Exchange of $CO_2$ between the atmosphere and ocean, and the biogeochemical consequences of this process, are governed by a series of equilibrium chemical reactions and parameters collectively known as the *marine carbonate system* (Millero, 2000). The core parameters are the substance contents of aqueous $CO_2$, the bicarbonate and carbonate ions formed by its hydration and dissociation ($HCO_3^-$ and $CO_3^{2-}$), and the sum of these three components (dissolved inorganic carbon, $T_C$); total alkalinity ($A_T$; Dickson, 1981); the fugacity, partial pressure, or dry-air mole fraction of $CO_2$ in seawater ($f_{CO_2}$, $p_{CO_2}$, or $x_{CO_2}$; Weiss, 1974); and pH (Dickson et al., 2015). If any valid pair of these parameters is known, plus auxiliary data including temperature, pressure, salinity and nutrient contents, then all the other parameters can be calculated (Park, 1969; Zeebe and Wolf-Gladrow, 2001).

Many research questions require solving the marine carbonate system from some measured or modelled pair of its parameters. Several software tools have been developed for this purpose, such that most scientific software environments and programming languages have a widely accepted marine carbonate system solver (Orr et al., 2015). However, there is not yet an established and fully functional tool for the popular scientific programming language Python, although partial solutions exist (e.g. Branson, 2018). Here, we present PyCO2SYS, a Python package designed to fill this capability gap and provide a robust platform for future developments in calculating marine chemical speciation. Being free and open source, and working across all major operating systems, a Python package is a highly accessible, desirable and useful tool.

As its name suggests, PyCO2SYS originates from the existing CO2SYS family of software. The original CO2SYS program for MS-DOS (Lewis and Wallace, 1998) has been further developed and 'translated', with implementations now available for Microsoft Excel (Pierrot et al., 2006; Orr et al., 2018; Pierrot et al., 2021) and MATLAB/GNU Octave (van Heuven et al., 2011; Xu et al., 2017; Orr et al., 2018; Sharp and Byrne, 2019; Sharp et al., 2020). PyCO2SYS was created as an as-close-as-possible translation of CO2SYS-MATLAB v2.0.5 (Orr et al., 2018), but we have since made several additional developments to it. Many of these developments involved reshaping the internal code into a more Pythonic style. These changes did not affect the calculations and so are not discussed further. Other developments added new functionality or made minor differences to the calculated results; these are documented and justified here.

As the original CO2SYS software is so well-established in the research field, we provide a relatively brief summary of the components of PyCO2SYS that are identical to CO2SYS-MATLAB in Sect. 2, before describing the areas where PyCO2SYS differs in more detail in Sect. 3. Equations that were inherited from CO2SYS-MATLAB or taken from the literature are generally reported in appendices rather than being reproduced in these sections. We go on to validate PyCO2SYS in Sect. 4 by examining its internal consistency and by comparing its calculations with another CO2SYS implementation. In Sect. 5, we discuss some nuances of solving the marine carbonate system that were explored during development and compare its computational speed with CO2SYS-MATLAB, before concluding with our perspectives on the outlook for PyCO2SYS and other related software.

## 2 Methods inherited from CO2SYS

The components of PyCO2SYS that have been inherited directly from CO2SYS-MATLAB v2.0.5 (Orr et al., 2018), with only the minimal changes needed to translate to Python plus aesthetic code restructuring, are described in this section.

### 2.1 Units and pH scales

The abundances of all solutes and total alkalinity provided as arguments to PyCO2SYS or returned from it as results are in units of $\mu mol \cdot kg^{-1}$, where kg is of the total solution. This means that they are neither concentrations nor molarity values, which are both per unit volume rather than mass, nor are they molality values, which are per kg of $H_2O$. Although sometimes referred to as *molinity*, the correct term is *substance content* (IUPAC, 1997), which we abbreviate to *content*.

Temperature is in °C and salinity is practical salinity, which is dimensionless (Millero et al., 2008).

Pressure is in dbar and represents the hydrostatic pressure exerted by the overlying water column, consistent with typical oceanographic conductivity-temperature-depth (CTD) measurement reporting. Atmospheric pressure is not included, so pressure is effectively zero in the laboratory and at the sea surface.

pH can be provided on the Free, Total, Seawater, and/or NBS scale, where $[H^+]$ is a substance content as noted above and thus in $mol \cdot kg$-solution$^{-1}$ (Appendix A; Zeebe and Wolf-Gladrow, 2001; Velo et al., 2010). In the results, pH is returned on all four of these scales.

### 2.2 Parameterisations and constants

A notable feature of all CO2SYS software is the variety of different parameterisation options to calculate the various equilibrium constants and some components' total contents from salinity, temperature and pressure. Which parameterisations the user selects can appreciably alter the results, so these choices should always be explicitly reported.

Some of these options also influence other, seemingly unrelated, parameters of other chemical systems. This is not widely appreciated, because this happens internally, hidden within the code. The most influential choice is for the carbonic acid dissociation constants, $K_1^*$ and $K_2^*$, for which there are 17 different options in PyCO2SYS (Table 1). We organise these options into three groups based on their effect on the 'hidden' internal parameterisations (Table 2):

1. Standard case. These are all identical, aside from their varying carbonic acid constants.

2. GEOSECS cases: GEOSECS-Takahashi and GEOSECS-Peng. GEOSECS-Peng treats phosphate differently with respect to its contribution to alkalinity, and this difference is reported in the results as the 'Peng correction'; see Lewis and Wallace (1998) for a more detailed explanation.

3. Freshwater case. Salinity and other total salt contents (ammonia, borate, calcium, fluoride, phosphate, silicate, sulfate and sulfide) are set to zero, irrespective of the user inputs.

Other internal settings are consistent across all cases (Table 3). These three cases have been present since the original CO2SYS for MS-DOS (Lewis and Wallace, 1998). That program included only options 1–8 for the carbonic acid dissociation

constants (Table 1), the others being published subsequent to its release. All subsequently added carbonic acid options follow the Standard case. While it is beyond the scope of this manuscript to judge the relative merits of the different options, in general 90  we recommend that one of the Standard cases be used unless there is a specific reason for doing otherwise.

In addition to the carbonic acid equilibria, the user has multiple parameterisation options for each of (i) the ratio between total borate and salinity, (ii) the bisulfate dissociation constant $K_{SO_4}^*$, and (iii) the hydrogen fluoride dissociation constant $K_{HF}^*$ (Tables 2 and 3). However, note that for (i), the user's choice is not respected in the GEOSECS cases, and neither (ii) nor (iii) is included at all in the Freshwater case (Table 2). It should also be noted that choices (ii) and (iii) affect pH scale conversions, 95  including of equilibrium constants, which can have a small (but practically negligible) effect on the results.

**Table 1.** Parameterisations of the dissociation constants of carbonic acid available in PyCO2SYS and corresponding implicit settings (Table 2).

| Option no. in PyCO2SYS | Carbonic acid constants | 'Other settings' case |
| --- | --- | --- |
| 1 | Roy et al. (1993) | Standard |
| 2 | Goyet and Poisson (1989) | Standard |
| 3 | Dickson and Millero (1987)[a] | Standard |
| 4 | Dickson and Millero (1987)[b] | Standard |
| 5 | Dickson and Millero (1987)[c] | Standard |
| 6 | Mehrbach et al. (1973) | GEOSECS-Takahashi |
| 7 | Mehrbach et al. (1973) | GEOSECS-Peng |
| 8 | Millero (1979)[d] | Freshwater |
| 9 | Cai and Wang (1998) | Standard |
| 10 | Lueker et al. (2000) | Standard |
| 11 | Mojica Prieto and Millero (2002) | Standard |
| 12 | Millero et al. (2002) | Standard |
| 13 | Millero et al. (2006) | Standard |
| 14 | Millero (2010) | Standard |
| 15 | Waters and Millero (2013)[e] | Standard |
| 16 | Sulpis et al. (2020) | Standard |
| 17 | Schockman and Byrne (2021) | Standard |

[a]Refit of Hansson (1973a, b) data. [b]Refit of Mehrbach et al. (1973) data. [c]Refit of Hansson (1973a, b) and Mehrbach et al. (1973) data. [d]Constants for zero-salinity freshwater. [e]Including the corrections of Waters et al. (2014).

Equilibrium constants in PyCO2SYS are all stoichiometric rather than thermodynamic and thus denoted with $K^*$. This means that they represent the equilibrium balance of solute substance contents, not of their chemical activities. They are evaluated as follows:

**Table 2.** Parameterisations that vary depending on the case of the selected carbonic acid constants (Table 1). $P$ = pressure.

| Setting | Standard | GEOSECS | Freshwater |
|---|---|---|---|
| Salinity | User-defined | User-defined | Zero |
| Total ammonia | User-defined | User-defined | Zero |
| Total borate | Uppström (1974) or[a] Lee et al. (2010) | Culkin (1965) | Zero |
| Total calcium | Riley and Tongudai (1967) | Culkin (1965) | Zero |
| Total fluoride | Riley (1965) | Riley (1965) | Zero |
| Total silicate | User-defined | User-defined | Zero |
| Total sulfate | Morris and Riley (1966) | Morris and Riley (1966) | Zero |
| Total phosphate | User-defined | User-defined[b] | Zero |
| Total sulfide | User-defined | User-defined | Zero |
| $K_1^*$ and $K_2^*$ $P$ effects | Millero (1995) | Takahashi et al. (1982) | Millero (1983) |
| $K_{H_2O}^*$ value | Millero (1995) | Millero (1979) | Millero (1979) |
| $K_{H_2O}^*$ $P$ effect | Millero (1995) | Millero (1995) | Millero (1983) |
| $K_B^*$ value | Dickson (1990b) | Li et al. (1969) | — |
| $K_B^*$ $P$ effect | Millero (1979) | Edmond and Gieskes (1970) | — |
| $K_P^*$ value[c] | Yao and Millero (1995) | Kester and Pytkowicz (1967) | — |
| $K_P^*$ $P$ effect[c] | Millero (1983) | Millero (1983) | — |
| $K_{Si}^*$ value | Yao and Millero (1995) | Sillén et al. (1964) | — |
| $K_{Si}^*$ $P$ effect | Millero (1995)[d] | Millero (1995)[d] | — |
| $K_{sp}^*$(calcite) value | Millero (1983) | Ingle (1975) | — |
| $K_{sp}^*$(calcite) $P$ effect | Ingle (1975) | Takahashi et al. (1982) | — |
| $K_{sp}^*$(aragonite) value | Millero (1983) | Ingle et al. (1973) | — |
| $K_{sp}^*$(aragonite) $P$ effect | Ingle (1975) | Takahashi et al. (1982) | — |
| Fugacity factor | Weiss (1974) | $1^e$ | Weiss (1974) |

[a] Depending on user input. [b] In GEOSECS-Takahashi, phosphate is not included in the definition of total alkalinity; in GEOSECS-Peng, phosphate is included, though the contribution of each species to alkalinity is determined incorrectly, based on charge rather than a zero-level of protons at p$K$ 4.5. [c] Includes all dissociation constants for this system: $K_{P1}^*$, $K_{P2}^*$ and $K_{P3}^*$ (Appendix B). [d] Copies the pressure correction for boric acid. [e] A constant value of 1 is used in this case, i.e. $p_{CO_2} = f_{CO_2}$.

1. Calculated on the pH scale reported in the literature, as a function of temperature and salinity, at zero in-water pressure;

2. Converted to the Seawater pH scale (Appendix A);

3. Corrected to the in situ pressure;

4. Converted to the pH scale indicated by the user's input (Appendix A).

**Table 3.** Parameterisations that (except where noted) are not influenced by the case of the selected carbonic acid constants (Table 1).

| Setting | References |
|---|---|
| $K_{SO_4}^*$ [a] | Khoo et al. (1977), Dickson (1990a), or Waters and Millero (2013)[b]; $P$ correction follows Millero (1995) |
| $K_{HF}^*$ [a] | Dickson and Riley (1979) or Perez and Fraga (1987)[c]; $P$ correction follows Millero (1995) |
| $K_{NH_3}^*$ | Clegg and Whitfield (1995); $P$ correction follows Millero (1995) |
| $K_{H_2S}^*$ | Yao and Millero (1995); $P$ correction follows Millero (1995) |
| $H^+$ activity coefficient | Takahashi et al. (1982), except for GEOSECS-Peng, which uses Peng et al. (1987) |
| Humidity correction | Weiss and Price (1980) |
| $CO_2$ solubility ($K_0^*$) | Weiss (1974) |

[a] As selected by the user. [b] Including the corrections of Waters et al. (2014). [c] This option was written into the code for CO2SYS-MATLAB v2.0.5 and other versions, but commented out and therefore not directly usable. It is available in CO2SYS-MATLAB v3.2.0.

There are some exceptions to the evaluation steps listed above. First, the pH scale conversions (steps 2 and 4) are not applied to $K_{SO_4}^*$, $K_{HF}^*$, $K_{sp}^*$(calcite), $K_{sp}^*$(aragonite), or $K_0^*$. For $K_{SO_4}^*$ and $K_{HF}^*$, this is because these constants always remain on the Free pH scale. The other constants in this list are for equilibria that do not directly involve $H^+$ and are therefore independent of the pH scale. Second, no pressure correction (step 3) is applied to the $CO_2$ solubility factor $K_0^*$ (Weiss, 1974). This value, and calculations of $f_{CO_2}$, $p_{CO_2}$ and $x_{CO_2}$, are thus valid only for the surface ocean (Sect. 5.3).

In PyCO2SYS, the user can also specify their own values for any or all of the equilibrium constants or total salt contents. Any values specified in this way are used as-is throughout PyCO2SYS: no pH scale or pressure corrections are applied, so it is left to the user to ensure that the values are provided on the appropriate pH scale and at the relevant temperature and pressure.

## 2.3 Input and output conditions

A useful feature of all CO2SYS software that nonetheless can cause confusion is calculations at 'input' and 'output' conditions, where 'conditions' refers to temperature and pressure. There is an unhelpful overlap of nomenclature, with 'input' and 'output' used firstly in a programming context to refer to arguments that are passed into functions and returned from them as results, and secondly in a measurement context where they refer to the temperatures and pressures under which the known parameter pairs are provided and at which results are to be calculated. For clarity, we therefore use the terms 'arguments' and 'results' in the programming context, while 'input' and 'output' always refer to the measurement context. Thus we provide values at both input and output conditions as arguments to PyCO2SYS and we receive calculations at both input and output conditions as results from the program.

Input and output conditions are used when measurements were conducted at a different temperature and/or pressure from what the sample would experience in situ, or to evaluate the effect of changing these conditions on the solution chemistry. All core carbonate system parameters except for $A_T$ and $T_C$ are temperature- and pressure-sensitive, so the values of other measured arguments and calculated results may differ between the input and output conditions. For example, measurements

might be conducted in a laboratory at 25 °C on samples that were collected from several kilometres' depth in the ocean at sub-zero temperatures. In this case, we would provide the measurement conditions (i.e. temperature and pressure in the laboratory) as input arguments, and the environmental conditions (i.e. temperature and pressure in the ocean) as output arguments. The corresponding output-condition results from PyCO2SYS then represent the true state of the sample in situ in its environment. The input-condition results are of less environmental interest but may be useful for quality-control purposes.

If calculations are conducted using only in situ values, for example from model output or where the temperature and pressure corrections have already been applied, then output-condition arguments need not be supplied. Results are then calculated only under the input conditions, for computational efficiency.

## 2.4 Solving the marine carbonate system

We refer to the parameters from which PyCO2SYS can solve the marine carbonate system as the 'core' marine carbonate system parameters. These are $A_\mathrm{T}$, $T_\mathrm{C}$, pH (on any scale), $p_{\mathrm{CO_2}}$, $f_{\mathrm{CO_2}}$, $x_{\mathrm{CO_2}}$, $[\mathrm{CO_2(aq)}]$, $[\mathrm{HCO_3^-}]$ and $[\mathrm{CO_3^{2-}}]$. Any pair of these can be provided, except for two of $p_{\mathrm{CO_2}}$, $f_{\mathrm{CO_2}}$, $x_{\mathrm{CO_2}}$ and $[\mathrm{CO_2(aq)}]$, which would not be valid because these are all directly proportional to each other at a given temperature, salinity, and atmospheric pressure.

To calculate its results (Fig. 1), PyCO2SYS first determines the unknown core parameters from whichever pair is provided by the user, under the input conditions (Appendix C). The parameter pairs that require an iterative solver to find pH (i.e. $A_\mathrm{T}$ plus $T_\mathrm{C}$ or one of its components) are solved using a scheme that has been updated from previous versions of CO2SYS (Sect. 3.1). The $A_\mathrm{T}$ and $T_\mathrm{C}$ provided or determined under the input conditions are then used to solve the core marine carbonate system again under the output conditions, if these have been provided. This is possible because both $A_\mathrm{T}$ and $T_\mathrm{C}$ are unaffected by temperature and pressure changes.

Other properties of interest are subsequently calculated from whichever core parameters are most convenient under both input and (if provided) output conditions. These properties include all the individual components of alkalinity (Appendix B), calcite and aragonite saturation states (Appendix D), and various chemical buffer factors (Sect. 3.3.4).

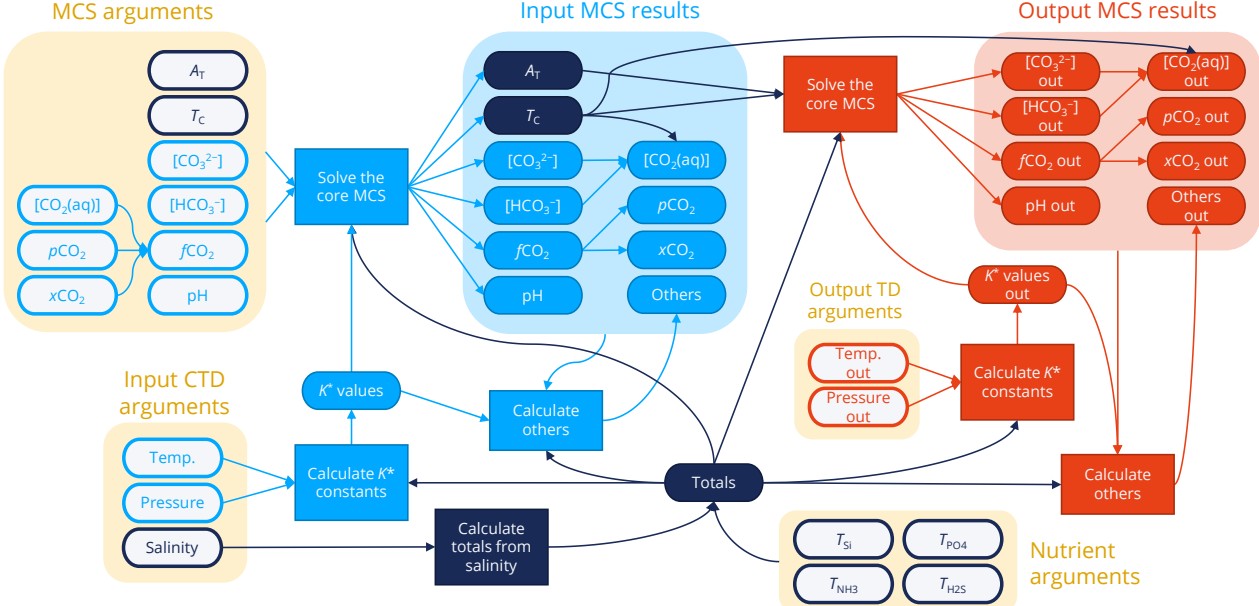

**Figure 1.** Overview of the process by which PyCO2SYS and other CO2SYS implementations solve the marine carbonate system (MCS) and calculate other results. Arguments provided by the user are shown as open symbols on a yellow background, while calculations and results use filled symbols. Components under input conditions are shown in light blue, those under output conditions are in red towards the right, and components that are independent of input/output conditions are in dark blue. Any pair of the parameters in the 'MCS arguments' box at the top left can be provided, noting that only one of $[CO_2(aq)]$, $p_{CO_2}$, $f_{CO_2}$ or $x_{CO_2}$ may be included in a pair. Coupled with user-provided nutrients, total salts calculated from salinity ('Totals'), and stoichiometric dissociation constants calculated from salinity and input temperature and pressure ('$K^*$ values'), all core MCS parameters are determined ('Input MCS results') from the known pair (Appendix C). Other results (e.g. carbonate mineral saturation states, buffer factors) are then calculated from the results under input conditions ('Others'). If the user provides output-condition temperature and/or pressure values, then the dissociation constants are recalculated under these new conditions, the core MCS is solved again ('Output MCS results') from these updated constants ('$K^*$ values out'), the original 'Totals', and the now-known $A_T$ and $T_C$, which are independent of temperature and pressure. Finally, other results are calculated again from the output-condition results ('Others out').

# 3 New developments in PyCO2SYS

## 3.1 Solving the alkalinity-pH equation

### 3.1.1 Automatic differentiation

Solving the alkalinity-pH equation is a critical component of marine carbonate system modelling. Like other implementations of CO2SYS, PyCO2SYS uses the Newton-Raphson method. The general equation is

$$\text{pH}_{n+1} = \text{pH}_n - \frac{\Delta A_\text{T}(\text{pH}_n, v)}{\Delta A'_\text{T}(\text{pH}_n, v)} \tag{1}$$

where $A'_\text{T} = \mathrm{d}A_\text{T}/\mathrm{dpH}$ and

$$\Delta A_\text{T}(\text{pH}_n, v) = A_\text{T}(\text{pH}_n, v) - A_\text{T}(\text{known}) \tag{2}$$

in which $v$ is any of $T_\text{C}$, $f_{\text{CO}_2}$, $[\text{HCO}_3^-]$ or $[\text{CO}_3^{2-}]$. $A_\text{T}(\text{pH}_n, v)$ is determined as described in Sects. C2.1 (when $v$ is $T_\text{C}$) and C2.3–C2.5 (when $v$ is one of $f_{\text{CO}_2}$, $[\text{CO}_3^{2-}]$ or $[\text{HCO}_3^-]$).

Unlike other implementations of CO2SYS, the equations that determine the relative abundances of different chemical species as functions of pH and their total contents (Appendix B) appear only once in PyCO2SYS, in what we term the 'main chemical speciation function'. While this approach does not alter the calculated results, it does make the software more robust by reducing the opportunity for typographical errors when similar equations are repeated across the code.

The derivative term in Eq. (1) is evaluated from the main chemical speciation function using automatic differentiation, as implemented by the Python package Autograd (Maclaurin, 2016). Distinct from numerical or symbolic differentiation, the automatic approach breaks down the code to be differentiated into a sequence of individual arithmetic operations (addition, subtraction, etc.) and simple functions (logarithms, exponentials, etc.), then combines the derivatives of these components together using the chain rule. The overall differentials to arbitrary order of complicated functions can thus be evaluated efficiently and accurate to the computer's precision.

Through our approach, the effect of every component of alkalinity in the main chemical speciation equation is included in the derivative term in Eq. (1). In contrast, some other implementations of CO2SYS use simplified expressions that only include the contributions of carbonate, borate and water to the total alkalinity. Under typical open-ocean conditions, this makes little practical difference, because the simplified equations include the most important components of the seawater solution. However, including every modelled component does make the solver more robust for more unusual solution compositions.

Automatic differentiation is also used to evaluate chemical buffer factors, again ensuring that the influence of every modelled equilibrium system is accurately included. The calculated buffer factors are described in more detail in Sect. 3.3.4.

A further advantage of the automatic-differentiation approach is that if the main chemical speciation function is modified in the future, for example to include additional components of alkalinity, then these changes are automatically incorporated into all the alkalinity-pH solvers without needing to modify the various solver functions. In short, our approach ensures that PyCO2SYS calculations will remain internally consistent and reflect the influence of every solute and equilibrium modelled in the main chemical speciation function, even if this function is modified in the course of future development (Sect. 5.5).

### 3.1.2 Vectorised arguments and solver jumps

PyCO2SYS adjusts how to determine when the alkalinity-pH solver should stop solving for vectorised arguments. In CO2SYS-MATLAB v2.0.5, the solvers continue to iterate and update all values until the change in every element of the array satisfies the $\Delta$pH tolerance threshold ($10^{-4}$ in CO2SYS-MATLAB, $10^{-8}$ in PyCO2SYS). This means that a given set of arguments could return slightly different results depending on what data appears in the other, supposedly independent, elements of the argument arrays. Although negligible for all practical purposes, these differences are detectable in code validation exercises. In PyCO2SYS (and in CO2SYS-MATLAB v3.2.0; Sharp et al., 2020) this process has been changed such that each element stops being updated once it has reached the tolerance threshold, independent of the other elements.

The maximum solver jump — which constrains the greatest change in pH possible between solver iterations, thus helping to prevent overshoot — is implemented slightly differently in PyCO2SYS than in other CO2SYS programs. In CO2SYS-MATLAB, any $\Delta$pH values with a magnitude greater than 1 are halved. In PyCO2SYS, the same applies, but any $\Delta$pH values with a magnitude still greater than 1 after halving are decreased to 1 (while preserving the sign). This has negligible effect on calculations but it is sometimes detectable in intercomparisons.

### 3.1.3 pH scale conversions

PyCO2SYS fixes a simplification in earlier CO2SYS implementations regarding how pH scales are converted within the main chemical speciation function. This simplification is noted in the programmer's comments in the relevant CO2SYS-MATLAB functions, carried through from the original MS-DOS implementation (Lewis and Wallace, 1998): "Though it is coded for H on the total pH scale, for the pH values occuring in seawater (pH > 6) it will be equally valid on any pH scale (H terms negligible) as long as the K Constants are on that scale."

In short, pH and the equilibrium constants are provided to these functions on the same pH scale as each other — except for $K_{SO_4}^*$ and $K_F^*$, which are always on the Free scale (Sect. 2.2). Calculations of all alkalinity components except $[HSO_4^-]$ and $[HF]$ have therefore always been correct. However, because $K_{SO_4}^*$ and $K_F^*$ are always on the Free scale, pH must be converted to this scale in order to determine the contributions of $[HSO_4^-]$ and $[HF]$ to total alkalinity. Other versions of CO2SYS prior to CO2SYS-MATLAB v3.2.0 (Sharp et al., 2020) and CO2SYS-Excel v3 (Pierrot et al., 2021) assume that the user-selected pH scale is Total, and thus apply the Total-to-Free scale conversion (Appendix A), regardless of what it the user-selected pH scale actually is.

This simplification makes negligible difference to calculations at typical seawater pH (because $[HSO_4^-]$ and $[HF]$ are each on the order of $10^{-10}$ $\mu$mol·kg$^{-1}$, relative to $A_T$ on the order of 2000 $\mu$mol·kg$^{-1}$) and then only when the user-selected pH scale is not Total. But, as implied in the original programmer's note, it can have a noticeable adverse effect under other conditions, such as the low pH values encountered during the acidimetric titrations of seawater used to measure $A_T$. In PyCO2SYS, CO2SYS-MATLAB v3.2.0, and CO2SYS-Excel v3, the correct conversion factor is used based on the user-selected pH scale.

## 3.2 Initial pH estimates

Like most iterative solvers, the Newton-Raphson method (Sect. 3.1) requires an initial pH value that is near to the true value in order to prevent overshoot and guarantee convergence to a root. Previous versions of CO2SYS used 8 as the initial pH estimate in every case. This works well for typical open-ocean seawater, but may be less appropriate in niche environments or when modelling acidimetric titrations. Munhoven (2013) found a better initial pH estimate for solving from known $A_T$ and $T_C$ by considering only the contributions of carbonate and borate species to $A_T$, simplifying the $A_T$ equation:

$$A_{CB} = [HCO_3^-] + 2[CO_3^{2-}] + [B(OH)_4^-] \tag{3}$$

Following Munhoven (2013) and as also implemented elsewhere (e.g. Orr and Epitalon, 2015), PyCO2SYS and CO2SYS-MATLAB v3.2.0 also take this approach (Appendix F). Furthermore, we have extended it to apply to the pH solvers that use one of the components of $T_C$ as the second known parameter, as follows. We note that these extensions are equivalent to those described and discussed in greater detail by Munhoven (2021), although they were added to the PyCO2SYS code in its v1.3.0 release (https://doi.org/10.5281/zenodo.3780139), before the publication of that study.

### 3.2.1 Solving from $A_T$ and $f_{CO_2}$

For clarity in the equations in this section, we abbreviate $[CO_2(aq)]$ as $s$, and $[H^+]$ as $h$. As noted in Appendix C1.2, the approach described here is also used for known parameter pairs of $A_T$ plus any of $p_{CO_2}$, $x_{CO_2}$ or $[CO_2(aq)]$.

First, $f_{CO_2}$ is converted to $s$ using Eq. (C5). Carbonate-borate alkalinity ($A_{CB}$) as a function of $s$ and $h$ is

$$A_{CB}(h, s) = \frac{K_1^* s(h + 2K_2^*)}{h^2} + \frac{K_B^* T_B}{h + K_B^*} \tag{4}$$

This can be rearranged into a third-order polynomial in $h$:

$$P_s(h, s) = h^3 + h^2 g_2(s) + h g_1(s) + g_0(s) = 0 \tag{5}$$

where

$$g_2(s) = K_B^* \left(1 - \frac{T_B}{A_{CB}}\right) - \frac{K_1^* s}{A_{CB}} \tag{6}$$

$$g_1(s) = \frac{(2K_2^* + K_B^*)K_1^* s}{-A_{CB}} \tag{7}$$

$$g_0(s) = \frac{2K_1^* K_2^* K_B^* s}{-A_{CB}} \tag{8}$$

Following an equivalent scheme to Munhoven (2013), the initial $h$ value is determined by

$$h_0(s) = \begin{cases} 10^{-3} & \text{for } A_T \leq 0 \\ h_{min} + \sqrt{-\dfrac{P_s(h_{min})}{\sqrt{g_2^2 - 3g_1}}} & \text{for } A_T > 0 \end{cases} \tag{9}$$

Negative $A_{\text{CB}}$ is impossible because both terms in Eq. (4) are always positive, so the equations given above cannot be applied
if $A_{\text{T}}$ is indeed negative (e.g. after the alkalinity end-point in an acidimetric titration). The default $h_0$ of $10^{-3}$ mol·kg$^{-1}$, corresponding to a pH of 3, is therefore used in this case. Otherwise, $h_{\text{min}}$ in Eq. (9) is found following Munhoven (2013):

$$
h_{\text{min}} = \begin{cases} (-g_2 + \sqrt{g_2^2 - 3g_1})/3 & \text{for } g_2 < 0 \\ -g_1/(g_2 + \sqrt{g_2^2 - 3g_1}) & \text{for } g_2 \geq 0 \end{cases}
\tag{10}
$$

When $A_{\text{T}}$ is positive, the square-rooted term $g_2^2 - 3g_1$ is always greater than zero, thus $h_{\text{min}}$ has a real value. However, there
is an additional constraint: $A_{\text{CB}}$ cannot be greater than $2T_{\text{C}} + T_{\text{B}}$ (Munhoven, 2013). If $A_{\text{T}}$ is actually greater than this limit,
then we use a default $h_0$ of $10^{-7}$ mol·kg$^{-1}$ instead (pH 7).

### 3.2.2  Solving from $A_{\text{T}}$ and $[\text{HCO}_3^-]$

For clarity in the equations in this section, we abbreviate $[\text{HCO}_3^-]$ as $b$, and $[\text{H}^+]$ as $h$.

Carbonate-borate alkalinity as a function of $b$ is

$$
A_{\text{CB}}(h,b) = b + \frac{2K_2^* b}{h} + \frac{K_{\text{B}}^* T_{\text{B}}}{h + K_{\text{B}}^*}
\tag{11}
$$

This can be rearranged into a second-order polynomial in $h$:

$$
P_b(h,b) = h^2 g_2(b) + h g_1(b) + g_1(b) = 0
\tag{12}
$$

where

$$
g_2(b) = b - A_{\text{CB}}
\tag{13}
$$
$$
g_1(b) = K_{\text{B}}^*(b + T_{\text{B}} - A_{\text{CB}}) + 2K_2^* b
\tag{14}
$$
$$
g_0(b) = 2K_2^* K_{\text{B}}^* b
\tag{15}
$$

The initial $h$ value is estimated following:

$$
h_0(b) = \begin{cases} \frac{-g_1 - \sqrt{g_1^2 - 4g_0 g_2}}{2g_2} & \text{for } b < A_{\text{T}} \\ 10^{-3} & \text{for } b \geq A_{\text{T}} \end{cases}
\tag{16}
$$

When $b < A_{\text{T}}$, the square-rooted term $g_1^2 - 4g_0 g_2$ is always positive and thus $h_0(b)$ has a real value. Otherwise, $b$ can only be
greater than $A_{\text{T}}$ if the negative components of $A_{\text{T}}$ such as $[\text{H}^+]$ are dominant, as happens at low pH. The default initial pH
estimate used by PyCO2SYS in that case is therefore 3.

### 3.2.3  Solving from $A_{\text{T}}$ and $[\text{CO}_3^{2-}]$

For clarity in the equations in this section, we abbreviate $[\text{CO}_3^{2-}]$ here as $c$, and $[\text{H}^+]$ as $h$.

Carbonate-borate alkalinity as a function of $c$ is:

$$A_{\mathrm{CB}}(h,c) = \frac{ch}{K_2^*} + 2c + \frac{K_{\mathrm{B}}^* T_{\mathrm{B}}}{h + K_{\mathrm{B}}^*} \tag{17}$$

This can be rearranged into a second-order polynomial in $h$:

$$P_c(h,c) = h^2 g_2(c) + h g_1(c) + g_0(c) = 0 \tag{18}$$

where

$$g_2(c) = c \tag{19}$$

$$g_1(c) = K_{\mathrm{B}}^* c + K_2^*(2c - A_{\mathrm{CB}}) \tag{20}$$

$$g_0(c) = K_2^* K_{\mathrm{B}}^*(2c + T_{\mathrm{B}} - A_{\mathrm{CB}}) \tag{21}$$

The initial $h$ value is estimated following:

$$h_0(c) = \begin{cases} \frac{-g_1 + \sqrt{g_1^2 - 4 g_0 g_2}}{2 g_2} & \text{for } A_{\mathrm{T}} > 2c + T_{\mathrm{B}} \\ 10^{-3} & \text{for } A_{\mathrm{T}} \le 2c + T_{\mathrm{B}} \end{cases} \tag{22}$$

When $2c + T_{\mathrm{B}} < A_{\mathrm{T}}$, the square-rooted term $g_1^2 - 4 g_0 g_2$ is always positive and thus $h_0(c)$ has a real value. Otherwise, $2c + T_{\mathrm{B}}$ can only be greater than $A_{\mathrm{T}}$ if the negative components of $A_{\mathrm{T}}$ such as $[\mathrm{H}^+]$ are dominant, as happens at low pH. The default
initial pH estimate used by PyCO2SYS in that case is therefore 3.

### 3.3   New calculations, components and constants

### 3.3.1   Additional alkalinity components

The contributions of ammonia and bisulfide to alkalinity (Cai et al., 2017; Xu et al., 2017) plus the ability to solve from carbonate and/or bicarbonate ion content have been added in collaboration with Sharp et al. (2020) to ensure consistency
between PyCO2SYS and CO2SYS-MATLAB v3.2.0. However, the GEOSECS alkalinity definition did not account for these species, so if using one of the GEOSECS options for the carbonic acid constants (Table 1) then the user should be sure to set their total contents to zero for GEOSECS-compatible results. If values are provided, then they will be included in the alkalinity equation just as for the non-GEOSECS cases.

The total substance contents and stoichiometric dissociation constants for up to two additional acid-base systems that con-
tribute to total alkalinity can be provided as arguments to PyCO2SYS and are part of its speciation model. The effects of these extra components are automatically incorporated into all PyCO2SYS calculations, including the iterative pH solvers (Sect. 3.1), buffer factors (Sect. 3.3.4), and uncertainty propagation (Sect. 3.6). These extra components are modelled following Sharp and Byrne (2020), as described in Appendix B11. No corrections of any sort (e.g. for pressure or pH scale; Sect. 2.2) are made to the dissociation constants for these user-defined additional components within PyCO2SYS; the user must ensure that they are
already suitable for the conditions being analysed and on the user-indicated pH scale.

### 3.3.2 Gas constant

Previous versions of CO2SYS used an old value for the universal gas constant ($R$) of 8.31451 J·mol$^{-1}$·K$^{-1}$. PyCO2SYS uses the 2018 CODATA recommended value by default instead (i.e. 8.314462618 J·mol$^{-1}$·K$^{-1}$), consistent with CO2SYS-MATLAB v3.2.0 and CO2SYS-Excel v3. This has a minor effect on conversions between $p_{CO_2}$, $f_{CO_2}$ and $x_{CO_2}$ (less than 10$^{-4}$ %), as well as on the pressure corrections for the equilibrium constants (less than 10$^{-3}$ % at 5000 dbar). It is detectable in comparisons with other versions of CO2SYS, but it is of no practical consequence.

### 3.3.3 Substrate:inhibitor ratio

Like CO2SYS-MATLAB v3.2.0, PyCO2SYS calculates the 'substrate:inhibitor ratio' of Bach (2015), which quantifies the balance between the availability of a substrate for calcification (i.e. $HCO_3^-$) and the inhibition of calcification by $H^+$ (Eq. (D2)).

### 3.3.4 Buffer factors

A buffer factor quantifies the sensitivity of a certain marine carbonate system parameter to a change in another parameter. Best known is the Revelle factor, which is the ratio of the fractional change in $p_{CO_2}$ corresponding to a fractional change in $T_C$ at constant $A_T$ (Revelle and Suess, 1957). Frankignoulle (1994) derived a broader set of buffer factors for the marine carbonate system, quantifying the responses of several different parameters to changes in $T_C$ and $A_T$; these were later rediscovered by Egleston et al. (2010) and further extended by Hagens and Middelburg (2016). PyCO2SYS calculates the buffer factors of Egleston et al. (2010) and uses the nomenclature of that manuscript.

Closely related to these buffer factors, Frankignoulle et al. (1994) introduced the factor $\psi$, which quantifies the change in $T_C$ required to return to the original seawater $p_{CO_2}$ after the action of calcification (which reduces $A_T$ and $T_C$ in a 2:1 ratio) or $CaCO_3$ dissolution (the reverse). Humphreys et al. (2018) introduced the 'isocapnic quotient' ($Q$), which is the ratio of $A_T$ to $T_C$ change that does not affect seawater $p_{CO_2}$, thus generalising the concept of $\psi$ for application to all biogeochemical processes that affect $A_T$ and $T_C$ (denoted $\phi$). PyCO2SYS calculates both $\psi$ and $Q$, the latter of which can be used to calculate $\phi$ for any biogeochemical process (Humphreys et al., 2018).

PyCO2SYS offers two independent ways to evaluate the various buffer factors of the marine carbonate system: with explicit equations and by automatic differentiation. The latter is used by default.

The 'explicit' approach follows equations reported in the literature (Frankignoulle et al., 1994; Egleston et al., 2010; Humphreys et al., 2018), noting that the typographical errors in Egleston et al. (2010) identified in several studies (e.g. Orr, 2011; Álvarez et al., 2014; Richier et al., 2018; Orr et al., 2018) have been corrected. In general, these equations do not include the effect of species beyond the carbonate, borate, and water contributions to total alkalinity, except that the buffer factors of Egleston et al. (2010) were extended to include phosphate and silicate effects by Orr et al. (2018).

The 'automatic' approach uses automatic differentiation to find the derivative necessary to evaluate each buffer factor. The appropriate derivatives are taken from the functions that calculate a third carbonate system parameter from a known pair (Appendix C). All species modelled in the main chemical speciation function are therefore included, including any extra

alkalinity components (Sect. 3.3.1), and typographical errors from the literature cannot influence these calculations. The details of the derivatives used are provided in Appendix E.

Of the buffer factors, only the Revelle factor was included in previous versions of CO2SYS. It was evaluated using finite central-difference derivatives, which is replicated as the 'explicit' option in PyCO2SYS (with the corrections described in Appendix G). However, as for all other buffer factors, the Revelle factor calculation uses automatic differentiation by default. To calculate the Revelle factor using a mathematically equivalent approach to the 'explicit' calculation of the other buffer factors, one could calculate $\gamma_{T_C}$ of Egleston et al. (2010) (see Appendix E1) with the explicit approach and then use Eq. (E7).

### 3.3.5 Atmospheric pressure

For conversions between $p_{CO_2}$, $f_{CO_2}$ and $x_{CO_2}$, atmospheric pressure is assumed to be 1 atm by default, and it remains fixed at this value in CO2SYS-MATLAB and CO2SYS-Excel. However, in PyCO2SYS, the user can also specify a value other than 1 atm, if necessary. Different values can be provided for input and output conditions (Sect. 2.3).

Atmospheric pressure can have a non-negligible effect on calculations in some regions: for example, over much of the Southern Ocean, atmospheric pressure is typically 3 % lower than the global mean, corresponding to a 10 µatm reduction in $p_{CO_2}$ and $f_{CO_2}$ relative to the values calculated at 1 atm (Orr et al., 2017).

This optional argument is only intended for modelling the effects of variations in atmospheric pressure on samples from the surface ocean or in the laboratory. It is not suitable for determining interior ocean $p_{CO_2}$, $f_{CO_2}$ and $x_{CO_2}$ values that are corrected for the pressure of the overlying water column. This separate issue is discussed further in Sect. 5.3.

### 3.4 No-solve modes

As well as solving from a pair of parameters, PyCO2SYS can be run with one or no marine carbonate system parameter arguments.

If no parameters are provided, then PyCO2SYS returns all the equilibrium constants and total salt contents that are calculated from temperature, pressure, and salinity (Sect. 2.2), without actually using these to do any further computations.

If one parameter is provided, then the results that can be computed with that parameter alone are returned. This applies to pH, $p_{CO_2}$, $f_{CO_2}$, $x_{CO_2}$, and $[CO_2(aq)]$, as follows.

pH can be converted between the different scales without knowledge of a second carbonate system parameter. Therefore if pH alone is provided to PyCO2SYS, it is converted to every pH scale under the input conditions (Appendix A). Conversion to a different temperature and/or pressure does require solving the carbonate system (Fig. 1), so output-condition values are not calculated.

Seawater $p_{CO_2}$, $f_{CO_2}$, $x_{CO_2}$, and $[CO_2(aq)]$ can also be interconverted without knowledge of a second carbonate system parameter (Appendix C1.2). Therefore if any of these parameters alone are provided to PyCO2SYS, all the others are calculated under the input conditions. If an output-condition temperature is provided, then $p_{CO_2}$ is also adjusted to the new temperature following Takahashi et al. (2009), and all others in this set of parameters are calculated under output conditions from the new $p_{CO_2}$ value.

### 3.5 Multidimensional arguments

All arguments to PyCO2SYS, including settings, can be multidimensional. A combination of scalar and multidimensional arguments can be provided, with the latter formatted as NumPy `ndarrays` (Harris et al., 2020). Results that depend only on scalar arguments are themselves scalar, while results depending on multidimensional inputs are 'broadcasted' into consistently shaped arrays (Fig. 2). The code is optimised to efficiently compute across multidimensional arrays following the approach of CO2SYS-MATLAB since its v1.1 (van Heuven et al., 2011). However, all multidimensional arrays in CO2SYS-MATLAB are flattened into one-dimensional vectors and returned in the results in that same format.

## (a) One-dimensional arguments

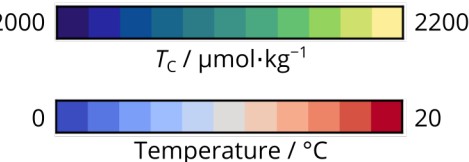

## (b) One-dimensional results

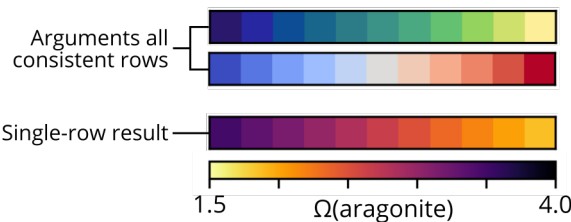

## (c) Multidimensional results

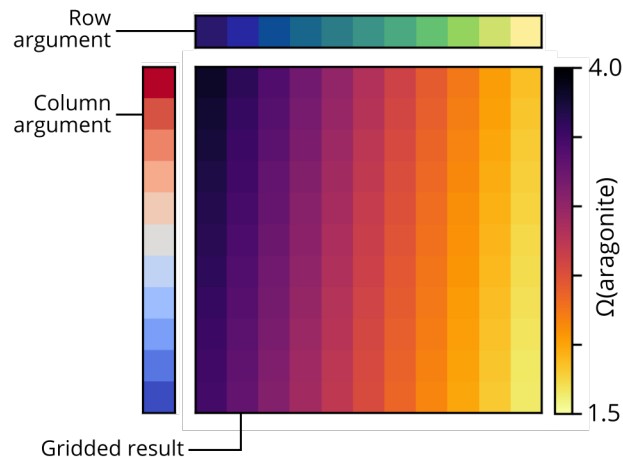

**Figure 2.** Schematic representation of broadcasting array shapes with NumPy in PyCO2SYS. (a) Two of the arguments to PyCO2SYS are provided as arrays, each containing 11 different values for $T_C$ and temperature. Other arguments could be similarly shaped vectors or single scalar values. (b) If the array arguments were all provided as one-dimensional rows, then the calculated results (e.g. aragonite saturation state) would also be one-dimensional rows. Each element of the results array corresponds to the element in the same position in each argument array. For scalar arguments, the same value is used across each result array. (c) If the array arguments are provided as a mixture of rows and columns, then the results are calculated on a broadcasted grid including every combination of the arguments' elements. The same principle applies to arguments and results of arbitrarily higher dimensionality.

### 3.6 Uncertainty propagation

Propagating the uncertainty in an argument through to a result requires knowing the derivative of the result with respect to the argument. Uncertainty propagation is available for a subset of the arguments in the original MS-DOS CO2SYS (Lewis and Wallace, 1998) and was added to the Excel and MATLAB implementations more recently (Orr et al., 2018). However, while much of the code to solve the marine carbonate system in PyCO2SYS has been directly inherited from CO2SYS-MATLAB, its implementation of uncertainty propagation differs.

PyCO2SYS evaluates the derivatives using a finite forward-difference approach. We use finite differences rather than automatic differentiation here because the latter, while possible, is computationally inefficient to apply over the entire PyCO2SYS program. We use forward- rather than central-difference derivatives because the former can be safely evaluated at zero for variables where negative values are impossible (e.g. salinity). The derivative of a result $r$ with respect to an argument $a$ is calculated thus:

$$\frac{\partial r(a)}{\partial a} \approx \frac{r(a + \Delta a) - r(a)}{\Delta a} \tag{23}$$

The value of $\Delta a$ is fixed for each argument (Appendix H). Different values for different arguments are necessary because some arguments can differ by over 20 orders of magnitude from others. If $\Delta a$ is too large, then the derivative may be inaccurate because the equations governing the marine carbonate system are non-linear, but if $\Delta a$ is too small, then the derivative may be inaccurate due to the limitations of solver tolerance and computer precision. We therefore tested a range of $\Delta a$ values for each variable under typical open-ocean conditions and selected an appropriate value between these extremes (e.g. Fig. 3). The full list of $\Delta a$ values is provided in Table H1.

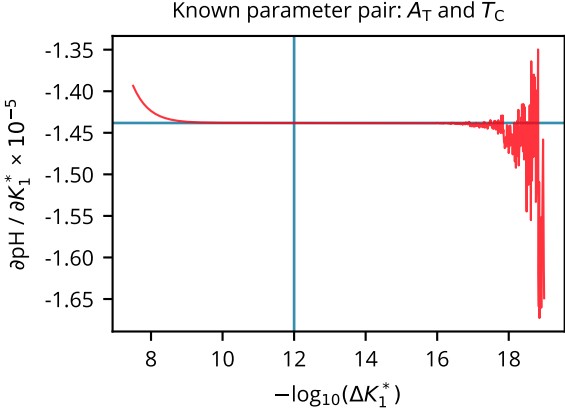

**Figure 3.** An example figure used to select a suitable $\Delta a$ value for uncertainty propagation, in this case for $\Delta K_1^*$. pH was calculated from $A_T$ and $T_C$ and then the value of $K_1^*$ was incremented by the amounts shown on the horizontal axis; the vertical axis is for the corresponding gradient calculated from the pH response, shown by the red curve. The perpendicular blue lines show the $\Delta a$ value selected in this case (i.e. $10^{-12}$), which falls within the flat section towards the centre of the figure. To the left of this (i.e. at higher $\Delta a$), the upwards curvature of the red line is due to non-linearity, while the erratic deviations to the right (i.e. at lower $\Delta a$) are due to solver tolerance and computer precision limitations.

PyCO2SYS can conveniently obtain derivatives of all its results with respect to all of its arguments and also with respect to all parameters that are normally calculated internally from temperature, pressure and/or salinity, such as equilibrium constants and total salt contents.

The derivatives are calculated by a function that wraps the entire PyCO2SYS program, rather than by adding extra internal variables that keep track of the effects of differences in to the arguments, as has been implemented elsewhere (e.g. Orr et al., 2018). The PyCO2SYS approach means that if the main program is producing valid results, then the derivatives can also be considered reliable without needing to verify some separate calculation mechanism.

To determine the overall uncertainty in each result, the uncertainty components from different arguments are combined using

$$\sigma^2(r) = \sum_i \left(\frac{\partial r}{\partial a_i}\right)^2 \sigma^2(a_i) \tag{24}$$

where $\sigma$ is the uncertainty as a standard deviation (thus $\sigma^2$ is a variance). However, Eq. (24) is only valid if the uncertainties in all arguments are independent from each other. Propagation of co-varying uncertainties can still be carried out with PyCO2SYS, because as noted above, the derivative of any result with respect to any argument can be calculated. The user can therefore assemble the Jacobian matrix of partial derivatives needed to propagate any arbitrary set of co-varying argument uncertainties through to any result (JCGM, 2008).

## 4    Validation

There are no 'certified' results of marine carbonate system calculations against which software like PyCO2SYS can be validated. But we can test its internal consistency and we can compare its results with the calculations of other programs and values reported in the literature.

PyCO2SYS is developed and hosted on GitHub (https://github.com/mvdh7/PyCO2SYS), with releases archived on Zenodo (Humphreys et al., 2021). Every validation test described in this section is built into PyCO2SYS's test suite, therefore these tests are executed automatically by GitHub's continuous integration service every time the code is updated. Were any test to fail, an email report would be sent to us, the developers, and the failure displayed publicly in a badge on the GitHub repository's public web page (Fig. 4). Updates to PyCO2SYS are made in a developmental branch of the repository and the tests must all pass before these changes may be incorporated into the main branch and publicly released in a new version. All validation tests described below were run with PyCO2SYS v1.8.0 (https://doi.org/10.5281/zenodo.5602840), but these protocols should ensure that the quantitative statements made here will hold true as the code continues to be developed.

For all versions of PyCO2SYS up to v1.8.0, the test suite runs on Python v3.7, 3.8 and 3.9. Other versions of Python may also work, but are untested.

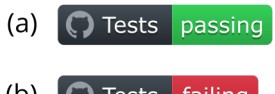

**Figure 4.** The status badge for the validation tests, publicly visible at PyCO2SYS's GitHub repository (https://github.com/mvdh7/PyCO2SYS), when the current version of the code (a) passes every test or (b) fails any test.

### 4.1    Internal consistency

#### 4.1.1    Round-robin test

In a 'round-robin' test, we first determine all of the core carbonate system parameters from one pair, and then solve the system again using every possible pair of determined parameters. Under typical seawater conditions, we find the same results for every parameter pair, to within better than the tolerance of the iterative pH solvers (i.e. $10^{-8}$ in pH). The maximum absolute difference in each parameter across all possible input pair combinations is acceptably small (Table 4).

#### 4.1.2    Buffer factors

If we include only the solution components that appear in the 'explicit' equations for the buffer factors (i.e. zero nutrients and total salts, except for $T_B$) then we can compare these results with the 'automatic' values (Sect. 3.3.4). Under a range of typical seawater conditions, we find that the differences between these two calculation approaches are totally negligible: on the order of $10^{-12}$ % for the Egleston et al. (2010) buffers; $10^{-9}$ % for $\psi$ and $Q$; and $10^{-7}$ % for the Revelle factor. The

**Table 4.** Results of an example round-robin test with PyCO2SYS with default parameterisation options. Other conditions: salinity = 33, temperature = 22 °C, pressure = 1234 dbar, total silicate = 10 $\mu mol \cdot kg^{-1}$, total phosphate = 1 $\mu mol \cdot kg^{-1}$, total ammonia = 2 $\mu mol \cdot kg^{-1}$, total sulfide = 3 $\mu mol \cdot kg^{-1}$. The pH-solver tolerance in PyCO2SYS is $10^{-8}$ in terms of pH.

| Parameter | Value | Maximum absolute difference |
|---|---|---|
| $A_T$ / $\mu mol \cdot kg^{-1}$ | 2300.0 | $5.91 \cdot 10^{-11}$ |
| $T_C$ / $\mu mol \cdot kg^{-1}$ | 2100.0 | $5.55 \cdot 10^{-11}$ |
| $pH_T$ | 7.871 | $1.15 \cdot 10^{-14}$ |
| $p_{CO_2}$ / $\mu atm$ | 572.6 | $1.51 \cdot 10^{-11}$ |
| $f_{CO_2}$ / $\mu atm$ | 570.7 | $1.50 \cdot 10^{-11}$ |
| $x_{CO_2}$ / $\mu atm$ | 587.7 | $1.55 \cdot 10^{-11}$ |
| $[CO_3^{2-}]$ / $\mu mol \cdot kg^{-1}$ | 143.8 | $3.81 \cdot 10^{-12}$ |
| $[HCO_3^-]$ / $\mu mol \cdot kg^{-1}$ | 1938.5 | $5.16 \cdot 10^{-11}$ |
| $[CO_2(aq)]$ / $\mu mol \cdot kg^{-1}$ | 17.7 | $6.54 \cdot 10^{-13}$ |

Revelle factor is less well-matched because its 'explicit' value is computed using a finite difference scheme (for consistency with CO2SYS-MATLAB), which is inherently less accurate than using a direct equation.

Typically, one would not set the total salt contents to zero when computing buffer factors with the default automatic approach. As a consequence, differences between the explicit and automatic buffer factors may be larger than described above, but still practically negligible: keeping nutrients at zero but using $T_{SO_4}$ and $T_F$ calculated from a salinity of 35, we find that the automatic buffer factors change such that their differences with the corresponding explicit buffer factors increase to the order of 0.01 %.

### 4.1.3 Uncertainty propagation simulations

The propagation of independent uncertainties using forward-difference derivatives (Sect. 3.6) is tested by comparison with Monte-Carlo simulations for all equilibrium constants and all known parameter pair combinations. In every case, the uncertainty determined from the simulations ($n = 10^4$) as a standard deviation is either within 3 % of the directly calculated value if the latter is non-zero, or negligibly small if it is zero (absolute value less than $10^{-10}$). The 3 % cutoff is relatively high because of the relatively small number of simulations; the cutoff can be reduced if a greater number of simulations is used, but then the computation time for the test suite becomes impractically long.

### 4.2 Comparison with other CO2SYS software

We used CO2SYS-MATLAB v2.0.5 (Orr et al., 2018) as the main alternative software to compare our results with. PyCO2SYS was originally created as an as-close-as-possible Python translation of this particular version, so any differences in the results

should be both understood and intentional. Its predecessor, CO2SYS-MATLAB v1.1 (van Heuven et al., 2011), was included in the software intercomparison study of Orr et al. (2015). Indeed, it was selected as the reference software to test the others against. CO2SYS-MATLAB v2.0.5 differs from v1.1 only in that it contains one additional parameterisation for the carbonic acid dissociation constants plus some extra internal variables associated with uncertainty propagation. Comparing PyCO2SYS with CO2SYS-MATLAB v2.0.5 therefore also shows PyCO2SYS's performance and reliability in the context of the wider set of software tested and discussed by Orr et al. (2015).

However, these CO2SYS-MATLAB versions do not permit solving with either carbonate or bicarbonate ion content as a known parameter, nor do they include ammonia or sulfide speciation. They also lack the parameterisations of Sulpis et al. (2020) and Schockman and Byrne (2021) for the carbonic acid dissociation constants (options 16 and 17 in Table 1), and the parameterisation of Waters and Millero (2013) for bisulfate dissociation (Table 3). We therefore also tested PyCO2SYS against CO2SYS-MATLAB v3.2.0 (Sharp et al., 2020), which does include all these options.

### 4.2.1 Temperature-salinity-pressure parameterisations

All equilibrium constants and total salt contents, calculated from salinity, temperature, and pressure, are virtually identical (absolute tolerance $10^{-12}$, relative tolerance $10^{-16}$, in p$K^*$ values or in $\mu mol \cdot kg^{-1}$) to those in both CO2SYS-MATLAB v2.0.5 and v3.2.0. These tests are run across a range of practical salinity from 0 to 50, temperature from $-1$ to 50 °C, and pressure from 0 to $10^5$ dbar, including values of exactly zero in every case. Every pH scale and parameterisation option is included (Tables 1 and 2).

### 4.2.2 Solving the marine carbonate system

If PyCO2SYS is adjusted to match CO2SYS-MATLAB v2.0.5, i.e.:

1. Approximate slopes are used for the pH solvers, including only carbonate-borate-water alkalinity, instead of using automatic differentiation to determine these exactly (Sect. 3.1.1);

2. pH solver tolerance is set to $10^{-4}$, instead of $10^{-8}$ (Sect. 3.1.2);

3. The original approach to prevent overshoot from too-great solver jumps in pH is used (Sect. 3.1.2);

4. The iterative pH solver continues updating all elements until all pH changes fall beneath the tolerance threshold (Sect. 3.1.2);

5. The pH-scale conversion simplification is reinstated (Sect. 3.1.3);

6. Initial pH guesses are always set to 8, instead of using our extended Munhoven (2013) approach (Sect. 3.2);

then the differences between PyCO2SYS and CO2SYS-MATLAB calculations are virtually zero (no greater than $10^{-10}$ %, excluding the Revelle factor as noted above). The Revelle factor is an exception, but this is due to minor errors in its encoding in CO2SYS-MATLAB (Appendix G). If we replicate these errors in PyCO2SYS, then we do return virtually identical Revelle factor values.

If the adjustments above, other than fixing the pH-scale conversion simplification, are not made, then the differences between PyCO2SYS and CO2SYS-MATLAB v2.0.5 are up to the order of $10^{-5}$ %: greater, but still negligible for all practical purposes.

Fixing the pH-scale conversion simplification too (Sect. 3.1.3) makes no difference to calculations where the user-defined input pH scale is Total, but causes discrepancies between PyCO2SYS and CO2SYS-MATLAB v2.0.5 of up to 50 % in the 'free' hydrogen ion content and $10^{-2}$ % in other results when other input pH scale options are selected. The differences are amplified at low pH, as the assumptions of the pH-scale conversion simplification do not hold (Sect. 3.1.3).

Repeating the exercise above for CO2SYS-MATLAB v3.2.0 has similar results, with differences negligible for all practical 470 purposes. Only adjustments 1, 2 and 3 from the list above need to be made to PyCO2SYS in this case. With PyCO2SYS fully adjusted to match CO2SYS-MATLAB v3.2.0, differences in calculated values are still mostly less than $10^{-10}$ %, and with one exception all less than $10^{-6}$ %. The exception, a difference still less than $10^{-3}$ %, is for the aqueous $CO_2$ content under a limited set of input conditions and only with the new known parameter pair combinations added since CO2SYS-MATLAB v2.0.5. It arises because there are several different ways to calculate $[CO_2(aq)]$: by difference from known $T_C$, $[HCO_3^-]$ and 475 $[CO_3^{2-}]$; from any one of these three variables, $[H^+]$, and $K_1^*$ and $K_2^*$ equilibrium constants using the equations in Appendix C (Sects. C2.6, C2.11 and C2.12); or from $f_{CO_2}$ or $p_{CO_2}$ and the $CO_2$ solubility constant ($K_0^*$). While these approaches are identical in theory, in practice they return different results due to the limitations of solver tolerance and floating point precision. PyCO2SYS and CO2SYS-MATLAB do not always use the same approach to calculate $[CO_2(aq)]$ in each situation (this also varies between CO2SYS-MATLAB versions), hence their greater — but still negligible — differences from each other. 480 Whatever the known parameter pair, PyCO2SYS always follows the principles that (i) the values of parameters provided as arguments by the user should never be overwritten with recalculations, and (ii) the final unknown from $T_C$, $[CO_3^{2-}]$, $[HCO_3^-]$ and $[CO_2(aq)]$ should always be calculated from the other three, by addition or by difference as appropriate.

### 4.2.3 Uncertainty propagation comparisons

PyCO2SYS reproduces all the derivatives reported by Orr et al. (2018) in their Tables 2 and 3 to within $10^{-3}$ % under the 485 same input conditions, and all the propagated uncertainties reported by Orr et al. (2018) in their Table 4 to within $10^{-4}$ %. We consider all these differences to be negligible.

Across all combinations of optional parameters, mean uncertainties in $A_T$, $T_C$, $p_{CO_2}$, $f_{CO_2}$, $[HCO_3^-]$, $[CO_3^{2-}]$, $[CO_2(aq)]$, $\Omega(\text{calcite})$, $\Omega(\text{aragonite})$ and $x_{CO_2}$ propagated from the standard values suggested by Orr et al. (2018) are within 0.5 % of the corresponding uncertainty values calculated with CO2SYS-MATLAB v3.2.0 under the same input conditions. Greater 490 differences in uncertainties calculated under output conditions arise because CO2SYS-MATLAB does not propagate the uncertainties from input-condition equilibrium constants through to output-condition results.

### 4.3 Simulated seawater titration

PyCO2SYS can be used to reproduce the closed-cell seawater titration datasets simulated by Dickson (1981). Each simulated dataset contains pH values for a seawater sample as it is titrated with incremental HCl additions across a pH range from 495 approximately 8 to 3.

Dickson (1981) specified exact values for all stoichiometric equilibrium constants. PyCO2SYS allows these to be provided, instead of them being calculated internally from temperature and salinity (2.2). The titration is then simulated by calculating how $A_T$ should change through the titration due to acid addition, accounting for dilution of $A_T$, $T_C$ and all other dissolved solutes by acid addition, and then solving the carbonate system for pH from the so-determined $A_T$ and $T_C$. On test here is the ability to solve for pH from known $A_T$ and $T_C$ across a wide range of pH and $A_T$ values, including negative $A_T$.

The first titration dataset, without phosphate, is reproduced perfectly by PyCO2SYS to the number of decimal places reported by Dickson (1981). The second titration, with 10 $\mu mol \cdot kg^{-1}$ of total phosphate included, is reproduced perfectly by PyCO2SYS with the exception of three values at different titrant masses:

- 0.45 g: pH either 6.5**88**221 (Dickson) or 6.5**99**221 (PyCO2SYS).

- 0.60 g: pH either 6.366**84**6 (Dickson) or 6.366**48**6 (PyCO2SYS).

- 1.25 g: pH either 5.549957 (Dickson) or 5.549951 (PyCO2SYS).

The other 48 data points in this titration agree perfectly. The noted discrepancies occur in non-consecutive data points and are therefore unlikely to all be associated with an error in a particular equilibrium. Coupled with the nature of the differences (underlined above), that is, one or two specific digits switched or replaced rather than the entire number being different, we conclude that these differences most likely represent minor typographical errors and therefore that PyCO2SYS does accurately reproduce these simulations in full.

## 5 Discussion

### 5.1 Initial pH estimates

The aim of our revised scheme for initial pH estimates, following Munhoven (2013), was to find values that were closer to the final solution across a wide range of pH, thus providing a more suitable starting point for the iterative solvers and thereby reducing the number of iterations required to converge at the solution.

We find that the initial pH estimates determined according to the scheme described in Sect. 3.2 do follow a similar pattern to the final solutions across wide ranges of argument values, including at the extremes where the initial-estimate equations become invalid and default pH values are used instead (Fig. 5). The number of iterations required to fall beneath the solver's tolerance threshold ($10^{-8}$ in pH) is also reduced, compared with the original approach of always using an initial pH of 8. Indeed, for typical ocean conditions we find that the iterative solver often does not alter the initial estimate at all. Suitable starting points for the iterative solvers are clearly being found.

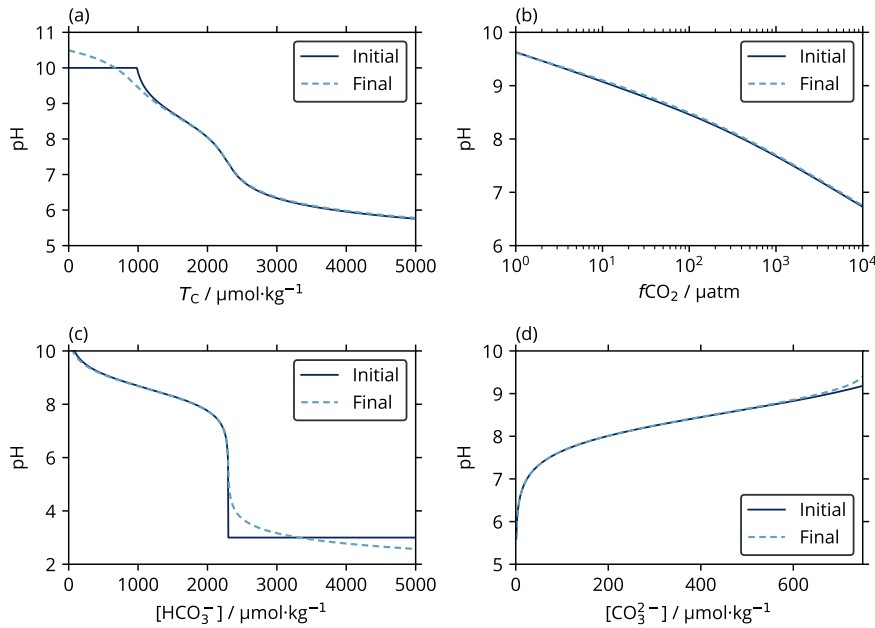

**Figure 5.** Initial estimates (solid lines) and final solutions (dashed lines) of pH from known parameter pairs of total alkalinity (2.3 mmol·kg$^{-1}$) with a range of values for (a) dissolved inorganic carbon ($T_C$), (b) aqueous $CO_2$ fugacity, (c) bicarbonate ion content, and (d) carbonate ion content. The initial estimates track the final solution very closely across the range of typical seawater conditions. This is expected, because these estimates were derived under the assumption that the carbonate and borate contributions are dominant in total alkalinity (Sect. 3.2), as is true for typical seawater. The default high and low pH values of 10 and 3 used where the initial estimate equations are not valid for the argument values (Eqs. (16) and (F6)) appear as flat sections in (a) and (c) respectively.

## 5.2 Parameter pairs with multiple solutions

It is not strictly true that the marine carbonate system can always be solved from any pair of its parameters. Some combinations have multiple solutions. For example, both the $A_T$-[CO$_3^{2-}$] and $T_C$-[HCO$_3^-$] pairs can correspond to two different pH values (Deffeyes, 1965; Zeebe and Wolf-Gladrow, 2001; Munhoven, 2021). In this section, we show how PyCO2SYS is designed to return the root corresponding to typical seawater. However, it is important to realise that these alternative pH values are real solutions that could be made up in the laboratory or be found in nature; they are not simply mathematical anomalies to be ignored. We therefore used PyCO2SYS to explore the compositions of these alternatives.

### 5.2.1 Total alkalinity and carbonate ion content

The iterative $A_T$-pH solvers can be thought of as working by evaluating $A_T$ at a sequence of different possible pH values until the pH that returns the true $A_T$ is found. This pH is known as the 'root' of the $A_T$-pH equation. The difference between the true $A_T$ and these estimates from pH is the 'residual' alkalinity, which is zero at the root. We find that the equations for initial

pH estimates and final pH values have very similar roots and similar residuals in the region around these roots (Fig. 6). This
similarity is why the initial pH estimates provide such suitable starting points for the final solvers.

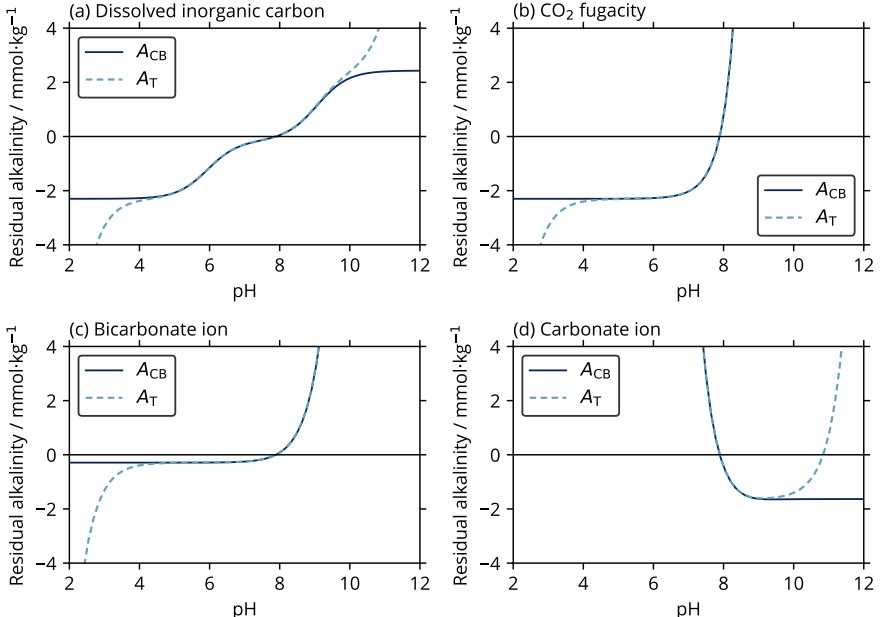

**Figure 6.** Residuals between known $A_T$ (2.3 mmol·kg$^{-1}$) and (i) carbonate-borate alkalinity (solid lines; $A_{CB}$) from Eqs. (F1), (4), (11)
and (17), and (ii) total alkalinity (dashed lines; $A_T$) from Eq. (B1), calculated across a range of pH, with a second known parameter of
(a) dissolved inorganic carbon (2.15 mmol·kg$^{-1}$), (b) $CO_2$ fugacity (600 µatm), (c) bicarbonate ion content (2011 µmol · kg$^{-1}$), and
(d) carbonate ion content (116 µmol · kg$^{-1}$), all at a salinity of 35 and temperature of 15 °C. Each possible pH value returns a different
residual alkalinity, and the true pH root is where the residual alkalinity is zero. Both the initial estimates and the final solutions find this
zero-residual pH root, using the $A_{CB}$ and $A_T$ equations respectively (Sects. 3.1.1 and 3.2). The similarity between the $A_{CB}$ and $A_T$ residual
curves, particularly around zero residual alkalinity, shows that the initial estimates provide excellent starting values for the subsequent
iterative solvers. In (d), the final iterative solver has two possible roots, where residual alkalinity is zero. However, the initial estimate has
only one root, corresponding to the lower-pH final root. This ensures that the final solver will always converge to the lower-pH root, which
is usually appropriate for the seawater system.

For the $A_T$-[$CO_3^{2-}$] parameter pair, there are generally two real pH roots and thus two possible equilibrium states of the
marine carbonate system (Fig. 6d). We used PyCO2SYS to conceptualise the two pH roots for the $A_T$-[$CO_3^{2-}$] parameter
pair, as follows. The lower-pH root corresponds to typical seawater: a relatively high-$T_C$ system, where bicarbonate ions are
the main component of $T_C$, and carbonate alkalinity ([$HCO_3^-$] + 2[$CO_3^{2-}$]) is the main component of $A_T$. The higher-pH
root corresponds to a low-$T_C$ system, where virtually all of $T_C$ is in the form of carbonate ion, and $A_T$ is dominated by
non-carbonate species (Fig. 7).

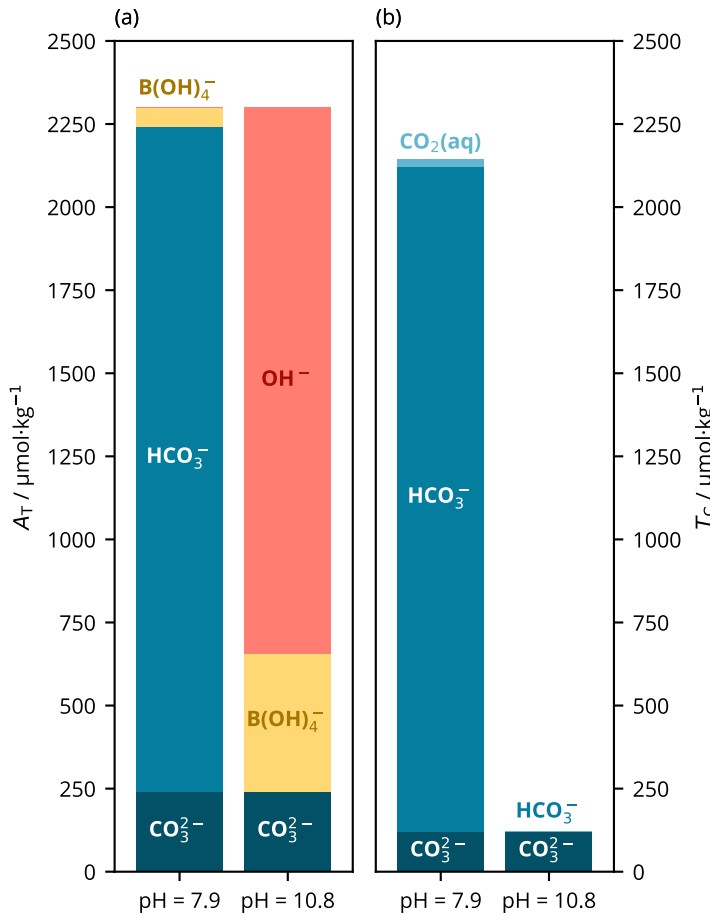

**Figure 7.** Main components of (a) total alkalinity ($A_T$) and (b) dissolved inorganic carbon ($T_C$) at the two possible pH roots for a known parameter pair of $A_T$ (2300 $\mu mol \cdot kg^{-1}$) and $[CO_3^{2-}]$ (120 $\mu mol \cdot kg^{-1}$). The low-pH root (left) represents typical seawater, with relatively high $T_C$ (2143 $\mu mol \cdot kg^{-1}$), and both $A_T$ and $T_C$ dominated by bicarbonate ion (HCO$_3^-$). The high-pH root (right) has the same $A_T$ and $[CO_3^{2-}]$, but $A_T$ is dominated by hydroxide (OH$^-$), and $T_C$ is much lower (122 $\mu mol \cdot kg^{-1}$) and comprised almost entirely of $CO_3^{2-}$. These calculations were carried out at 15 °C, with a practical salinity of 35 and zero nutrients. If nutrients were present, then like borate (B(OH)$_4^-$) they would have different contributions to $A_T$ at the different pH roots. pH is on the Total scale (Appendix A).

Which root the solver finds depends on the initial pH estimate and the residual alkalinity-pH slope at that point (Eq. (1)). This is an advantage of the improved initial pH estimates in PyCO2SYS when working with seawater and similar systems: the initial-estimate equation has only a single real root (Fig. 6d). Because the initial estimate is based on equations for a system that only includes carbonate and borate alkalinity (Sect. 3.2.3), the carbonate system contribution to total alkalinity will always dominate, so the single root of the initial estimate will coincide with the lower-pH true root, which is appropriate for seawater. The solver will thus more robustly find the correct root each time.

In typical open-ocean work this is largely academic: the true pH is typically around 8, and the higher root greater than 10, so a constant initial pH estimate of 8 would also return the correct root. But in more unusual environments, the new algorithm

introduced here could help ensure that the solver identifies the correct root. It is possible for the user to specify a different initial pH estimate, to control which root PyCO2SYS obtains (as we did to create Fig. 7).

### 5.2.2 Dissolved inorganic carbon and bicarbonate ion content

As noted previously (e.g. Zeebe and Wolf-Gladrow, 2001), there are also two possible pH solutions for the $T_C$-[$HCO_3^-$] parameter pair. We conceptualise these roots as follows: the remaining portion of $T_C$ not accounted for by $HCO_3^-$ is either

dominantly composed of $CO_3^{2-}$ if the solution's pH is closer to $pK_2^*$ (i.e. higher), or of $CO_2(aq)$ if the pH is closer to $pK_1^*$ (i.e. lower).

Solving from $T_C$ and [$HCO_3^-$] is more straightforward than from $A_T$ and [$CO_3^{2-}$] because the unknown pH can be determined from a second-order polynomial, which can be calculated directly using the quadratic formula, rather than needing to use an iterative solver. Here, the root found does not depend upon the value of some initial pH estimate. Instead, the quadratic

formula generates two possible roots, which must be chosen between. The usual approach, as advised by e.g. Zeebe and Wolf-Gladrow (2001), is to take the higher-pH root, and this is the default behaviour of PyCO2SYS. However, PyCO2SYS can be set to return the other root instead, which we used to illustrate the differing chemistry of the two possibilities (Fig. 8). Munhoven (2021) discusses root-selection strategy for this parameter pair combination in more detail.

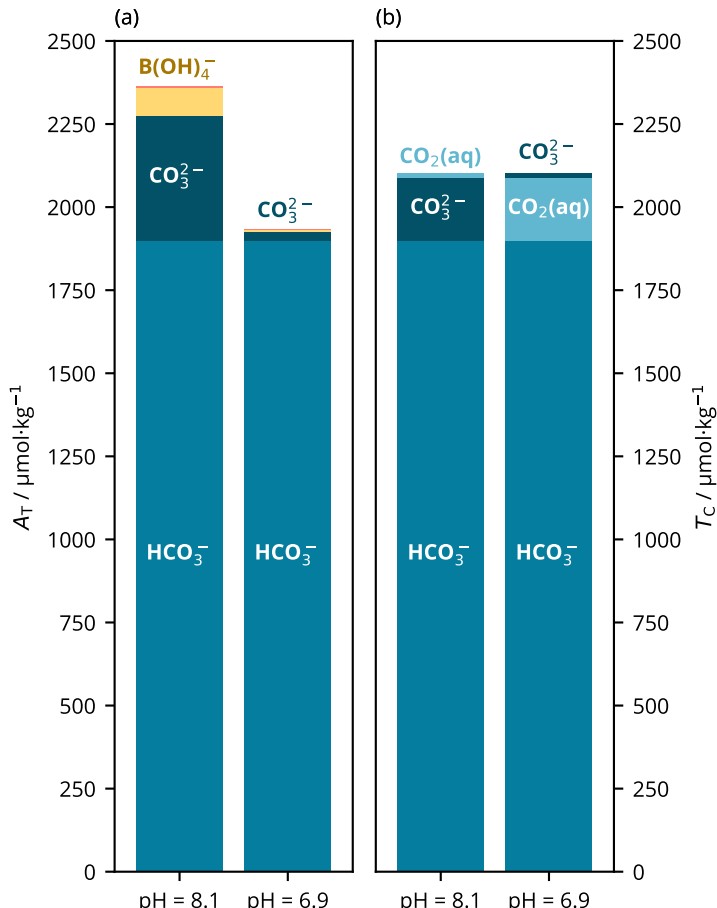

**Figure 8.** Main components of (a) total alkalinity ($A_T$) and (b) dissolved inorganic carbon ($T_C$) at the two possible pH roots for a known parameter pair of $T_C$ (2100 $\mu mol \cdot kg^{-1}$) and bicarbonate ion content ([$HCO_3^-$]; 1900 $\mu mol \cdot kg^{-1}$). The high-pH root (left) represents typical seawater, where most of the $T_C$ not accounted for by [$HCO_3^-$] is composed of [$CO_3^{2-}$]; $A_T$ is relatively high (2364 $\mu mol \cdot kg^{-1}$) and $f_{CO_2}$ low (331 $\mu atm$). In the low-pH root (right), the non-$HCO_3^-$ portion of $T_C$ is instead dominated by $CO_2(aq)$; alkalinity is lower (1932 $\mu mol \cdot kg^{-1}$) and $f_{CO_2}$ high (5008 $\mu atm$). These calculations were carried out at 15 °C, with a practical salinity of 35 and zero nutrients. pH is on the Total scale (Appendix A).

## 5.3 Pressure corrections for $p_{CO_2}$

In PyCO2SYS, $p_{CO_2}$ (and by extension, $f_{CO_2}$ and $x_{CO_2}$) is always evaluated at a total pressure near 1 atm; it is not corrected for the pressure of the overlying water column (Sect. 2.2). This approach is consistent with all existing implementations of CO2SYS. In practice, it means that these values represent the approximate $p_{CO_2}$ that seawater would have if it were brought to the surface ocean without changing the solution composition — 'approximate' because this calculation should use potential temperature, rather than in situ temperature, to retrieve the true value expected after adiabatic decompression (Orr and Epi-

talon, 2015). PyCO2SYS does not calculate potential temperature, but this could be provided by the user in place of in situ temperature.

Although a pressure correction for $p_{CO_2}$ (i.e. a pressure correction for $K_0^*$ and the fugacity factor; Appendix C1.2) is theoretically possible (Weiss, 1974; Orr and Epitalon, 2015), it could be argued that this is unnecessary. First, the vast majority of $p_{CO_2}$ measurements are carried out only at the surface ocean (e.g. Bakker et al., 2016), in part due to practical constraints of the 'gold-standard' equilibrator-based methodology. Second, the concept of $p_{CO_2}$ has utility only in the context of air-sea $CO_2$ exchange, which takes place only at the surface ocean.

However, recent developments in sensor technology are beginning to enable direct measurements of in situ $p_{CO_2}$ at depth in the ocean (Clarke et al., 2017). There is also growing interest in calculating in situ $p_{CO_2}$ values at depth for intercomparison exercises in which the marine carbonate system has been overdetermined by measuring more than two of its core parameters (e.g. Raimondi et al., 2019), and the relevant pressure correction is implemented in software tools such as seacarb and mocsy (Orr and Epitalon, 2015; Orr et al., 2015). Therefore, we do anticipate an increasing need for pressure-corrected $p_{CO_2}$ values, and while we have kept the approach in PyCO2SYS consistent with other CO2SYS software for now, we consider a robust implementation of these calculations to be an important target for future code development.

### 5.4 Computational speed

One does not choose to write code in Python for its computational speed. Therefore, while optimising performance was not ignored in developing PyCO2SYS, it was not a main focus. We compared the computational speed of PyCO2SYS against that of CO2SYS-MATLAB v3.2.0 across a few different tasks for reference purposes. We ran CO2SYS-MATLAB both in MATLAB itself (expensive, proprietary software) and in GNU Octave, a free and open source MATLAB clone.

The different tasks are described in the subsequent sections and the results are summarised in Table 5. Details of the computer and software used for testing are provided in Appendix I.

**Table 5.** Comparison of computational speed for various tasks with PyCO2SYS and CO2SYS-MATLAB running in both MATLAB and GNU Octave. Values shown are the mean $\pm$ standard deviation of 7 runs. The tasks are described in Sect. 5.4.

| Task | Python time / s | MATLAB time / s | GNU Octave time / s |
|---|---|---|---|
| All combinations | $0.95 \pm 0.04$ | $0.68 \pm 0.11$ | $0.64 \pm 0.01$ |
| GLODAP — input only | $23.8 \pm 0.3$ | $13.1 \pm 0.3$ | $16.5 \pm 0.9$ |
| GLODAP — input and output | $49.9 \pm 2.7$ | $13.1 \pm 0.3$ | $16.5 \pm 0.9$ |

Overall, the PyCO2SYS computation time has the same order of magnitude as CO2SYS-MATLAB, but it is generally somewhat slower. However, the difference is negligible in practice for relatively small datasets (up to about $10^5$ data points), but may become more noticeable in larger calculations. Potential future improvements to PyCO2SYS's computational speed are discussed in Sect. 5.5.

### 5.4.1 All combinations

The 'all combinations' task was the validation test described in Sect. 4.2.2, that is, a single call to the (Py)CO2SYS function that includes one calculation using every possible combination of parameter pair and optional setting (e.g. choices of parameterisations for the equilibrium constants): 40,800 data points. Both input and output conditions were computed.

CO2SYS-MATLAB completed this task in a very similar time in both MATLAB and GNU Octave with the latter slightly faster, and PyCO2SYS took about 1.5 times longer (Table 5). However, this difference would generally be negligible, as all three implementations of the test had an average run time of less than one second.

### 5.4.2 GLODAP

In this task, (Py)CO2SYS was run across the entire GLODAPv2.2021 Merged Master File (Lauvset et al., 2021) with $A_\text{T}$ and $T_\text{C}$ as the known parameter pair. This file contains a little over 1.3 million data points for each variable. The results in Table 5 show the mean and standard deviation of 7 runs in each case.

This calculation is an example where results would only be required under one set of temperature and pressure conditions, rather than needing to evaluate both input and output conditions. This allows PyCO2SYS to be used more efficiently, as it only calculates output-condition results if they are explicitly requested (Sect. 2.3), whereas CO2SYS-MATLAB always calculates its results at both input and output conditions.

The results in Table 5 show that CO2SYS-MATLAB running in MATLAB was the fastest, with GNU Octave taking longer by a factor of about 1.3. When calculating only under input conditions, PyCO2SYS took longer by a factor of about 1.8 than CO2SYS-MATLAB running in MATLAB, and by 3.8 if both the input- and output-condition calculations were carried out.

### 5.5 Outlook

The Autograd package that PyCO2SYS uses for automatic differentiation is still being maintained, and its most recent release (v1.3, July 2019) is stable, but it is no longer in active development. Its successor, JAX (Bradbury et al., 2018), has further benefits including 'just-in-time' code compilation and parallelisation. These features could speed up computation speed in PyCO2SYS, especially the components involving automatic differentiation, potentially by several orders of magnitude. However, JAX cannot currently run natively on the Microsoft Windows operating system, which would greatly restrict the usability of PyCO2SYS for the oceanographic research community. This limitation is due to JAX's dependence on the separate XLA (Accelerated Linear Algebra) compiler, rather than being an intrinsic issue with JAX itself. Should this compatibility issue be resolved in the future, we envision updating PyCO2SYS to use JAX instead of Autograd. This should be relatively straightfoward thanks to the close similarities between the API (application programming interface) of these packages.

As future developments are made to PyCO2SYS, we will aim to maintain consistency with other CO2SYS-family tools, but cannot guarantee that all new features or updates will be added simultaneously across all implementations. In practice, the workload required to achieve this is not currently feasible, and we would not wish to hold back development because of the

time required to replicate changes across multiple implementations. That said, the results should remain consistent enough that users can select which implementation to use based on their preferred software environment, rather than the other way around.

This ambition could also extend beyond the CO2SYS family of software. Independently developed tools for solving the marine carbonate system exist in other languages, such as seacarb in R (Gattuso et al., 2021) and mocsy in Fortran (Orr and Epitalon, 2015). These give sufficiently consistent results with each other that the selection of which tool to use does not affect scientific interpretation (Orr et al., 2015), and we have shown that PyCO2SYS is, and will remain, no exception. Even so, development and validation of PyCO2SYS so far has focused on comparisons with only CO2SYS-family software, for practical reasons. Now that the basis of PyCO2SYS is established, we would welcome more direct interaction with the groups developing these other tools, working towards a set of marine-carbonate-system-solving tools that return identical results regardless of the software platform. There can be a great advantage in having independent implementations led by different groups of researchers and developers. For example, this approach can help catch bugs and typographical errors, especially if each group extracts equations and parameterisations from the original literature instead of copying existing code. Working together, the groups would have a greater pool of knowledge and experience to identify errors in the literature (see e.g. Lewis and Wallace (1998), their Appendix A), which are often unpublished and known only through personal communications. But calculations must be regularly compared with each other if this advantage is to be realised.

Thanks largely to the efforts of Orr et al. (2018), many tools now have an uncertainty propogation capability, as does PyCO2SYS. However, we still lack meaningful and statistically equivalent estimates for the actual uncertainties in the equilibrium constants. The software therefore stands ahead of our knowledge: as more work is done to robustly quantify these uncertainties, the tools are already in place to propagate them through to all marine carbonate system calculations.

As development of PyCO2SYS continues, we do not anticipate changing its fundamental approach to solving the marine carbonate system, but we will try to incorporate the latest research, including keeping up-to-date with new parameterisations, for example of stoichiometric equilibrium constants (e.g. Sulpis et al., 2020; Schockman and Byrne, 2021). Integration with a speciation model that can determine the equilibrium constants based on chemical activities, rather than parameterising these based on salinity, is an area of interest (Turner et al., 2016), but would likely require such substantial changes as to constitute a separate software tool. We do envision further additions to the main chemical speciation function in PyCO2SYS, for example to better represent the impact of organic contributions to alkalinity (e.g. Cantrell et al., 1990; Muller and Bleie, 2008; Kuliński et al., 2014; Abril et al., 2015; Ulfsbo et al., 2015) — noting that a simplified representation of such extra components can already be modelled in PyCO2SYS (Sect. 3.3.1).

Through all these efforts, we aim to ensure that PyCO2SYS remains a reliable and comprehensive tool for analysing seawater chemistry, from samples and experiments in the laboratory through to the changing marine carbonate system across the global ocean.

*Code availability.* The current version of PyCO2SYS is freely available from its GitHub repository at https://github.com/mvdh7/PyCO2SYS under the GNU General Public License v3. Installation is recommended from the Python Package Index (PyPI) via `pip` and documenta-

tion is available online (https://PyCO2SYS.readthedocs.io). The exact version of PyCO2SYS used to produce the results discussed in this
paper (v1.8.0), including input data and scripts to run the model and perform all validation tests described here, is archived on Zenodo
(https://doi.org/10.5281/zenodo.5602840).

## Appendix A:  pH scales and conversions

The pH scales in PyCO2SYS are Free ($\text{pH}_F$), Total ($\text{pH}_T$), Seawater ($\text{pH}_S$) and NBS ($\text{pH}_N$), defined following e.g. Zeebe and
Wolf-Gladrow (2001) and Velo et al. (2010):

$$\text{pH}_F = -\log_{10}\{[\text{H}^+]\} \tag{A1}$$

$$\text{pH}_T = -\log_{10}\{[\text{H}^+](1+T_{\text{SO}_4}/K^*_{\text{SO}_4})\} \tag{A2}$$

$$\text{pH}_S = -\log_{10}\{[\text{H}^+](1+T_{\text{SO}_4}/K^*_{\text{SO}_4}+T_{\text{F}}/K^*_{\text{F}})\} \tag{A3}$$

$$\text{pH}_N = -\log_{10}\{[\text{H}^+](1+T_{\text{SO}_4}/K^*_{\text{SO}_4}+T_{\text{F}}/K^*_{\text{F}})\gamma_{\text{H}^+}\} \tag{A4}$$

where $\gamma_{\text{H}^+}$ is the chemical activity coefficient for $\text{H}^+$ (Table 3). Note that in PyCO2SYS, $[\text{H}^+]$ in all these definitions is
a substance content (Sect. 2.1). pH values and stoichiometric equilibrium constants ($K^*$) are thus converted between these
different pH scales using the following factors:

$$Y_F^T = 1 + T_{\text{SO}_4}/K^*_{\text{SO}_4} \; ; \; Y_T^F = 1/Y_F^T \tag{A5}$$

$$Y_F^S = 1 + T_{\text{SO}_4}/K^*_{\text{SO}_4} + T_{\text{F}}/K^*_{\text{F}} \; ; \; Y_S^F = 1/Y_F^S \tag{A6}$$

$$Y_S^N = \gamma_{\text{H}^+} \; ; \; Y_N^S = 1/Y_S^N \tag{A7}$$

where $\gamma_{\text{H}^+}$ is the hydrogen ion activity, calculated from temperature and salinity following either Peng et al. (1987) or Taka-
hashi et al. (1982) (see Table 3). The different scales are denoted by the subscript and superscript letters, with $F$ for Free, $T$
for Total, $S$ for Seawater and $N$ for NBS. To convert from any pH scale $A$ to any other pH scale $B$ using these factors:

$$\text{pH}_B = \text{pH}_A + \text{p}Y_A^B = \text{pH}_A - \log_{10}\left(Y_A^B\right) \tag{A8}$$

Alternatively and equivalently:

$$[\text{H}^+]_B = Y_A^B[\text{H}^+]_A \tag{A9}$$

The equations above are used in the same way to convert $K^*$ values between pH scales.

## Appendix B:  Total alkalinity and its components

Each equation here is written assuming that $[\text{H}^+]$ and all equilibrium constants ($K^*$) are supplied on the same pH scale as each
other.

 **B1 Total alkalinity**

Total alkalinity ($A_T$) is calculated as the sum of all its components (Dickson, 1981; Wolf-Gladrow et al., 2007; Sharp and Byrne, 2020):

$$A_T = A_w + A_C + A_B + A_P + A_{Si} + A_{NH_3} + A_{H_2S} + A_{SO_4} + A_F + A_\alpha + A_\beta \tag{B1}$$

Equations for all the individual alkalinity components ($A_C$, $A_B$, etc.) are given in the subsequent sections in terms of pH-independent total substance contents ($T_C$, $T_B$, etc.) and [$H^+$].

**B2 Water**

$$H_2O \rightleftharpoons OH^- + H^+ \; ; \; K_w^* = [OH^-][H^+] \tag{BR1}$$

$$A_w = [OH^-] - [H^+] = \frac{K_w^*}{[H^+]} - [H^+] \tag{B2}$$

**B3 Carbonic acid**

$$T_C = [CO_2(aq)] + [HCO_3^-] + [CO_3^{2-}] \tag{B3}$$

$$CO_2(aq) + H_2O \rightleftharpoons HCO_3^- + H^+ \; ; \; K_1^* = \frac{[HCO_3^-][H^+]}{[CO_2(aq)]} \tag{BR2}$$

$$HCO_3^- \rightleftharpoons CO_3^{2-} + H^+ \; ; \; K_2^* = \frac{[CO_3^{2-}][H^+]}{[HCO_3^-]} \tag{BR3}$$

$A_C$ can be expressed in terms of [$H^+$] and any of $T_C$, $f_{CO_2}$, [$HCO_3^-$] or [$CO_3^{2-}$]:

$$A_C = [HCO_3^-] + 2[CO_3^{2-}] \tag{B4}$$

$$A_C([H^+], T_C) = \frac{T_C K_1^* ([H^+] + 2K_2^*)}{K_1^* K_2^* + K_1^* [H^+] + [H^+]^2} \tag{B5}$$

$$A_C([H^+], f_{CO_2}) = \frac{f_{CO_2} K_0^* K_1^* ([H^+] + 2K_2^*)}{[H^+]^2} \tag{B6}$$

$$A_C([H^+], [HCO_3^-]) = [HCO_3^-] + \frac{2K_2^* [HCO_3^-]}{[H^+]} \tag{B7}$$

$$710 \quad A_\mathrm{C}([\mathrm{H}^+], [\mathrm{CO}_3^{2-}]) = \frac{[\mathrm{CO}_3^{2-}][\mathrm{H}^+]}{K_2^*} + 2[\mathrm{CO}_3^{2-}] \tag{B8}$$

Undissociated $\mathrm{H}_2\mathrm{CO}_3$ is considered negligible and thus not explicitly modelled, but rather implicitly included as part of the $[\mathrm{CO}_2(\mathrm{aq})]$ term (Zeebe and Wolf-Gladrow, 2001).

## B4  Boric acid

$$T_\mathrm{B} = [\mathrm{B(OH)}_3] + [\mathrm{B(OH)}_4^-] \tag{B9}$$

$$\mathrm{B(OH)}_3 + \mathrm{H}_2\mathrm{O} \rightleftharpoons \mathrm{B(OH)}_4^- + \mathrm{H}^+ \ ; \ K_\mathrm{B}^* = \frac{[\mathrm{B(OH)}_4^-][\mathrm{H}^+]}{[\mathrm{B(OH)}_3]} \tag{BR4}$$

$$A_\mathrm{B} = [\mathrm{B(OH)}_4^-] = \frac{T_\mathrm{B} K_\mathrm{B}^*}{K_\mathrm{B}^* + [\mathrm{H}^+]} \tag{B10}$$

## B5  Phosphoric acid

$$720 \quad T_\mathrm{P} = [\mathrm{H}_3\mathrm{PO}_4] + [\mathrm{H}_2\mathrm{PO}_4^-] + [\mathrm{HPO}_4^{2-}] + [\mathrm{PO}_4^{3-}] \tag{B11}$$

$$\mathrm{H}_3\mathrm{PO}_4 \rightleftharpoons \mathrm{H}_2\mathrm{PO}_4^- + \mathrm{H}^+ \ ; \ K_\mathrm{P1}^* = \frac{[\mathrm{H}_2\mathrm{PO}_4^-][\mathrm{H}^+]}{[\mathrm{H}_3\mathrm{PO}_4]} \tag{BR5}$$

$$\mathrm{H}_2\mathrm{PO}_4^- \rightleftharpoons \mathrm{HPO}_4^{2-} + \mathrm{H}^+ \ ; \ K_\mathrm{P2}^* = \frac{[\mathrm{HPO}_4^{2-}][\mathrm{H}^+]}{[\mathrm{H}_2\mathrm{PO}_4^-]} \tag{BR6}$$

$$\mathrm{HPO}_4^{2-} \rightleftharpoons \mathrm{PO}_4^{3-} + \mathrm{H}^+ \ ; \ K_\mathrm{P3}^* = \frac{[\mathrm{PO}_4^{3-}][\mathrm{H}^+]}{[\mathrm{HPO}_4^{2-}]} \tag{BR7}$$

$$A_\mathrm{P} = [\mathrm{HPO}_4^{2-}] + 2[\mathrm{PO}_4^{3-}] - [\mathrm{H}_3\mathrm{PO}_4] = \frac{T_\mathrm{P}(K_\mathrm{P1}^* K_\mathrm{P2}^* [\mathrm{H}^+] + 2 K_\mathrm{P1}^* K_\mathrm{P2}^* K_\mathrm{P3}^* - [\mathrm{H}^+]^3)}{K_\mathrm{P1}^* K_\mathrm{P2}^* K_\mathrm{P3}^* + K_\mathrm{P1}^* K_\mathrm{P2}^* [\mathrm{H}^+] + K_\mathrm{P1}^* [\mathrm{H}^+]^2 + [\mathrm{H}^+]^3} \tag{B12}$$

## B6  Orthosilicic acid

$$730 \quad T_\mathrm{Si} = [\mathrm{H}_4\mathrm{SiO}_4] + [\mathrm{H}_3\mathrm{SiO}_3^-] \tag{B13}$$

$$\mathrm{H}_4\mathrm{SiO}_4 \rightleftharpoons \mathrm{H}_3\mathrm{SiO}_4^- + \mathrm{H}^+ \ ; \ K_\mathrm{Si}^* = \frac{[\mathrm{H}_3\mathrm{SiO}_4^-][\mathrm{H}^+]}{[\mathrm{H}_4\mathrm{SiO}_4]} \tag{BR8}$$

$$A_{\text{Si}} = [\text{H}_3\text{SiO}_4^-] = \frac{T_{\text{Si}} K_{\text{Si}}^*}{K_{\text{Si}}^* + [\text{H}^+]} \tag{B14}$$

Further deprotonation of $\text{H}_3\text{SiO}_4^-$ is considered negligible and thus not modelled.

## B7   Ammonium

$$T_{\text{NH}_3} = [\text{NH}_3] + [\text{NH}_4^+] \tag{B15}$$

$$\text{NH}_4^+ \rightleftharpoons \text{NH}_3 + \text{H}^+ \; ; \; K_{\text{NH}_3}^* = \frac{[\text{NH}_3][\text{H}^+]}{[\text{NH}_4^+]} \tag{BR9}$$

$$A_{\text{NH}_3} = [\text{NH}_3] = \frac{T_{\text{NH}_3} K_{\text{NH}_3}^*}{K_{\text{NH}_3}^* + [\text{H}^+]} \tag{B16}$$

## B8   Sulfide

$$T_{\text{H}_2\text{S}} = [\text{H}_2\text{S}] + [\text{HS}^-] \tag{B17}$$

$\text{H}_2\text{S} \rightleftharpoons \text{HS}^- + \text{H}^+ \; ; \; K_{\text{H}_2\text{S}}^* = \frac{[\text{HS}^-][\text{H}^+]}{[\text{H}_2\text{S}]}$            (BR10)

$$A_{\text{H}_2\text{S}} = [\text{HS}^-] = \frac{T_{\text{H}_2\text{S}} K_{\text{H}_2\text{S}}^*}{K_{\text{H}_2\text{S}}^* + [\text{H}^+]} \tag{B18}$$

Further deprotonation of $\text{HS}^-$ is considered negligible and thus not modelled (Schoonen and Barnes, 1988).

## B9   Sulfate

$T_{\text{SO}_4} = [\text{HSO}_4^-] + [\text{SO}_4^{2-}]$            (B19)

$$\text{HSO}_4^- \rightleftharpoons \text{SO}_4^{2-} + \text{H}^+ \; ; \; K_{\text{SO}_4}^* = \frac{[\text{SO}_4^{2-}][\text{H}^+]}{[\text{HSO}_4^-]} \tag{BR11}$$

$$A_{\text{SO}_4} = -[\text{HSO}_4^-] = \frac{-T_{\text{SO}_4}}{1 + K_{\text{SO}_4}^*/[\text{H}^+]} \tag{B20}$$

Undissociated $\text{H}_2\text{SO}_4$ is considered negligible and thus not modelled.

## B10  Fluoride

$$T_F = [HF] + [F^-] \tag{B21}$$

$$HF \rightleftharpoons F^- + H^+ \; ; \; K_F^* = \frac{[F^-][H^+]}{[HF]} \tag{BR12}$$

$$A_F = -[HF] = \frac{-T_F}{1 + K_F^*/[H^+]} \tag{B22}$$

## B11  Arbitrary additional components

$$T_\alpha = [H\alpha] + [\alpha^-] \tag{B23}$$

$$H\alpha \rightleftharpoons \alpha^- + H^+ \; ; \; K_\alpha^* = \frac{[\alpha^-][H^+]}{[H\alpha]} \tag{BR13}$$

$$A_\alpha = \begin{cases} -[H\alpha] & \text{for } -\log_{10}(K_\alpha^*) \leq 4.5 \\ +[\alpha^-] & \text{for } -\log_{10}(K_\alpha^*) > 4.5 \end{cases} \tag{B24}$$

The reactions and equations for the second additional component $\beta$ and its alkalinity contribution $A_\beta$ are identical to those given for $\alpha$ above. PyCO2SYS automatically determines how to modify the alkalinity equation following Eq. (B24) based on

the user-provided $K_\alpha^*$ and $K_\beta^*$ values, with a zero-level of protons corresponding to a p$K^*$ of 4.5 (Wolf-Gladrow et al., 2007).

Though the definition of alkalinity (Dickson, 1981) states that species are separated into proton acceptors and donors based on their dissociation constant at zero ionic strength and 25 °C, we use the user-defined dissociation constants at the given conditions because one cannot convert arbitrary dissociation constants to their alkalinity-relevant values. Interpretations of results when arbitrary components are supplied to PyCO2SYS with p$K^*$ values close to 4.5 should consider this nuance.

## Appendix C:  Solving the core marine carbonate system

Here, we lay out all the equations that are used to convert between different carbonate system parameters in PyCO2SYS. These follow long-established approaches from the literature (Zeebe and Wolf-Gladrow, 2001; Dickson et al., 2007). The equations are organised based on which parameter pair is initially known.

### C1 General considerations

 ### C1.1 pH to [H$^+$] conversions

As the stoichiometric equilibrium constants are converted to the user-specified pH scale, i.e. consistent with the pH values, pH and [H$^+$] are interconverted in the equations throughout this section using

$$pH = -\log_{10}[H^+] \tag{C1}$$

regardless of which pH scale is being used.

### C1.2 Known $p_{CO_2}$, $x_{CO_2}$ or [CO$_2$(aq)]

If one of $p_{CO_2}$, $x_{CO_2}$ or [CO$_2$(aq)] is in the known parameter pair, then its values are first converted to $f_{CO_2}$ as follows.

For known $p_{CO_2}$:

$$f_{CO_2} = G p_{CO_2} \tag{C2}$$

where $G$ is the fugacity factor (Table 2), typically near 0.997.

For known $x_{CO_2}$:

$$f_{CO_2} = G P_v x_{CO_2} \tag{C3}$$

where $P_v$ is the humidity correction (Table 3):

$$P_v = P_a - p_w \tag{C4}$$

in which $P_a$ is total atmospheric pressure (assumed to be 1 atm unless a different value is provided by the user) and $p_w$ is the water vapour pressure (Weiss and Price, 1980).

For known [CO$_2$(aq)]:

$$f_{CO_2} = \frac{[CO_2(aq)]}{K_0^*} \tag{C5}$$

where $K_0^*$ is the solubility factor for CO$_2$ (Table 3).

The calculation steps given below for $f_{CO_2}$ are then followed to solve the core marine carbonate system. Afterwards, $p_{CO_2}$, $x_{CO_2}$ and [CO$_2$(aq)] are calculated where they were not in the original known parameter pair: $p_{CO_2}$ and $x_{CO_2}$ are calculated using Eqs. (C2) and (C3), while [CO$_2$(aq)] is calculated by difference using the definition of $T_C$ in Eq. (B3).

### C2 Solving routines

### C2.1 From $A_T$ and $T_C$

An initial pH estimate is determined as described in Appendix F. The estimate is then revised using the iterative approach of Sect. 3.1, in which the $A_T(pH_n, v)$ term in Eq. (2) is calculated from Eq. (B1) for $A_T$ substituting in Eq. (B5) for the $A_C$ term. Equation (2) is the automatically differentiated with respect to pH to obtain the $\Delta A_T'$ term in Eq. (1).

The components of $T_C$ are then calculated from $T_C$ and the final pH value:

$$f_{CO_2} = \frac{T_C[H^+]^2}{K_0^*([H^+]^2 + K_1^*[H^+] + K_1^*K_2^*)} \tag{C6}$$

$$[HCO_3^-] = \frac{T_C K_1^*[H^+]}{[H^+]^2 + K_1^*[H^+] + K_1^*K_2^*} \tag{C7}$$

$$[CO_3^{2-}] = \frac{T_C K_1^*K_2^*}{[H^+]^2 + K_1^*[H^+] + K_1^*K_2^*} \tag{C8}$$

### C2.2 From $A_T$ and pH

First, we determine $A_C$ from known $A_T$ and pH by using Eq. (B1). $T_C$ is then calculated from $A_C$:

$$T_C = \frac{A_C([H^+]^2 + K_1^*[H^+] + K_1^*K_2^*)}{K_1^*([H^+] + 2K_2^*)} \tag{C9}$$

The components of $T_C$ are then calculated from $T_C$ and pH using Eqs. (C6), (C7) and (C8).

There is an upper limit on pH for each given $A_T$ value, above which negative $A_C$ would be required to balance Eq. (B1). PyCO2SYS prints a warning if such an impossible pairing is used and returns NaN (not a number) for $T_C$ (and all other results calculated from it) instead of a negative value.

### C2.3 From $A_T$ and $f_{CO_2}$

An initial pH estimate is determined as described in Appendix F. The estimate is then revised using the iterative approach of Sect. 3.1, in which the $A_T(pH_n, v)$ term in Eq. (2) is calculated from Eq. (B1) for $A_T$ substituting in Eq. (B6) for the $A_C$ term. Equation (2) is the automatically differentiated with respect to pH to obtain the $\Delta A_T'$ term in Eq. (1).

$T_C$ is then calculated from $A_T$ and pH following Sect. C2.2, and its remaining unknown components with Eqs. (C7) and (C8).

### C2.4 From $A_T$ and $[CO_3^{2-}]$

An initial pH estimate is determined as described in Appendix F. The estimate is then revised using the iterative approach of Sect. 3.1, in which the $A_T(pH_n, v)$ term in Eq. (2) is calculated from Eq. (B1) for $A_T$ substituting in Eq. (B8) for the $A_C$ term. Equation (2) is the automatically differentiated with respect to pH to obtain the $\Delta A_T'$ term in Eq. (1). The lower of the two pH roots is returned by default, as discussed in Sect. 5.2.2.

$T_C$ is then calculated from $A_T$ and pH following Sect. C2.2, and its remaining unknown components with Eqs. (C6) and (C7).

### C2.5 From $A_T$ and $[HCO_3^-]$

An initial pH estimate is determined as described in Appendix F. The estimate is then revised using the iterative approach of Sect. 3.1, in which the $A_T(pH_n, v)$ term in Eq. (2) is calculated from Eq. (B1) for $A_T$ substituting in Eq. (B7) for the $A_C$ term. Equation (2) is the automatically differentiated with respect to pH to obtain the $\Delta A_T'$ term in Eq. (1).

$T_C$ is then calculated from $A_T$ and pH following Sect. C2.2, and its remaining unknown components with Eqs. (C6) and (C8).

### C2.6   From $T_C$ and pH

First, $A_T$ is calculated from $T_C$ and pH using Eq. (B1). The components of $T_C$ are then calculated from $T_C$ and pH using Eqs. (C6), (C7) and (C8).

### C2.7   From $T_C$ and $f_{CO_2}$

First, pH is calculated from $T_C$ and $f_{CO_2}$ using

$$[H^+] = \frac{K_1^* r + \sqrt{(K_1^* r)^2 + 4(1-r) K_1^* K_2^* r}}{2(1-r)} \tag{C10}$$

where

$$r = K_0^* \cdot f_{CO_2} / T_C \tag{C11}$$

$A_T$ and the remaining unknown components of $T_C$ are then calculated from $T_C$ and pH using Eqs. (B1), (C7) and (C8).

### 845   C2.8   From $T_C$ and $[CO_3^{2-}]$

First, pH is calculated from $T_C$ and $[CO_3^{2-}]$ using

$$[H^+] = \frac{-K_1^* + \sqrt{K_1^{*2} - 4K_1^* K_2^* (1 - T_C/[CO_3^{2-}])}}{2} \tag{C12}$$

$A_T$ and the remaining unknown components of $T_C$ are then calculated from $T_C$ and pH using Eqs. (B1), (C6) and (C7).

### C2.9   From $T_C$ and $[HCO_3^-]$

First, pH is calculated from $T_C$ and $[HCO_3^-]$ using

$$[H^+] = \frac{T_C - [HCO_3^-] - \sqrt{([HCO_3^-] - T_C)^2 - 4[HCO_3^-]^2 K_2^*/K_1^*}}{2[HCO_3^-]/K_1^*} \tag{C13}$$

$A_T$ and the remaining unknown components of $T_C$ are then calculated from $T_C$ and pH using Eqs. (B1), (C6) and (C8).

### C2.10   From pH and $f_{CO_2}$

First, $T_C$ is calculated from pH and $f_{CO_2}$ using

$$855 \quad T_C = \frac{K_0^* \cdot f_{CO_2} ([H^+]^2 + K_1^*[H^+] + K_1^* K_2^*)}{[H^+]^2} \tag{C14}$$

$A_T$ and the remaining unknown components of $T_C$ are then calculated from $T_C$ and pH using Eqs. (B1), (C7) and (C8).

### C2.11   From pH and $[CO_3^{2-}]$

First, $f_{CO_2}$ is calculated from pH and $[CO_3^{2-}]$ using

$$f_{CO_2} = \frac{[CO_3^{2-}][H^+]^2}{K_0^* K_1^* K_2^*} \tag{C15}$$

$T_C$ is then calculated from pH and $f_{CO_2}$ using Eq. (C14). Finally, $A_T$ and $[HCO_3^-]$ are calculated from $T_C$ and pH using Eqs. (B1) and (C7) respectively.

### C2.12   From pH and $[HCO_3^-]$

First, $T_C$ is calculated from pH and $[HCO_3^-]$ using

$$T_C = [HCO_3^-]\left(1 + \frac{[H^+]}{K_1^*} + \frac{K_2^*}{[H^+]}\right) \tag{C16}$$

$A_T$ and the remaining unknown components of $T_C$ are then calculated from $T_C$ and pH using Eqs. (B1), (C6) and (C8).

### C2.13   From $f_{CO_2}$ and $[CO_3^{2-}]$

First, pH is calculated from $f_{CO_2}$ and $[CO_3^{2-}]$ using

$$[H^+] = \sqrt{\frac{K_0^* K_1^* K_2^* \cdot f_{CO_2}}{[CO_3^{2-}]}} \tag{C17}$$

$T_C$ is then calculated from pH and $f_{CO_2}$ using Eq. (C14). Finally, $A_T$ and $[HCO_3^-]$ are calculated from $T_C$ and pH using Eqs. (B1) and (C7) respectively.

### C2.14   From $f_{CO_2}$ and $[HCO_3^-]$

First, $[CO_3^{2-}]$ is calculated from $f_{CO_2}$ and $[HCO_3^-]$ using

$$[CO_3^{2-}] = \frac{[HCO_3^-]^2 K_2^*}{K_0^* K_1^* \cdot f_{CO_2}} \tag{C18}$$

pH is then calculated from $f_{CO_2}$ and $[CO_3^{2-}]$ using Eq. (C17). Next, $T_C$ is calculated from pH and $f_{CO_2}$ using Eq. (C14). Finally, $A_T$ is calculated from $T_C$ and pH using Eq. (B1).

### C2.15   From $[CO_3^{2-}]$ and $[HCO_3^-]$

First, $f_{CO_2}$ is calculated from $[CO_3^{2-}]$ and $[HCO_3^-]$ using

$$f_{CO_2} = \frac{[HCO_3^-]^2 K_2^*}{K_0^* K_1^* [CO_3^{2-}]} \tag{C19}$$

pH is then calculated from $f_{CO_2}$ and $[CO_3^{2-}]$ using Eq. (C17). Next, $T_C$ is calculated from pH and $f_{CO_2}$ using Eq. (C14). Finally, $A_T$ is calculated from $T_C$ and pH using Eq. (B1).

## Appendix D: Other marine carbonate system variables

Calcite and aragonite saturation states ($\Omega$) are calculated from the definition:

$$\Omega = \frac{[\text{Ca}^{2+}][\text{CO}_3^{2-}]}{K_{\text{sp}}^*} \tag{D1}$$

where $K_{\text{sp}}^*$ is the solubility product, a function of salinity, temperature and pressure that is different for each mineral (Table 2).

The 'substrate:inhibitor ratio' of Bach (2015) is calculated from the bicarbonate and free hydrogen ion contents:

$$\text{SIR} = \frac{[\text{HCO}_3^-]}{[\text{H}^+]} \tag{D2}$$

Note that in Eq. (D2), the $[\text{H}^+]$ term is always calculated on the Free pH scale of Eq. (A1).

## Appendix E: Buffer factors with automatic differentiation

### E1    Buffer factors of Egleston et al. (2010)

To evaluate the buffer factors of Egleston et al. (2010) with automatic differentiation (AD), we first evaluated the following partial differentials (with the subscripted variable held constant):

     – $(\partial T_\text{C}/\partial \text{pH})_{A_\text{T}}$ by AD of Eq. (C9) with respect to pH;

     – $(\partial A_\text{T}/\partial \text{pH})_{T_\text{C}}$ by AD of Eq. (B1), substituting $A_\text{C}$ by Eq. (B5), with respect to pH;

     – $(\partial \ln[\text{CO}_2(\text{aq})]/\partial \text{pH})_{T_\text{C}}$ by taking the natural log of the product of $K_0^*$ and Eq. (C6), then AD with respect to pH;

– $(\partial \ln[\text{CO}_2(\text{aq})]/\partial \text{pH})_{A_\text{T}}$ by taking the natural log of the product of $K_0^*$ and Eq. (C6), substituting $T_\text{C}$ by Eq. (9), then AD with respect to pH.

The buffer factors $\gamma_{T_\text{C}}$, $\gamma_{A_\text{T}}$, $\beta_{T_\text{C}}$ and $\beta_{A_\text{T}}$ are thus defined (Egleston et al., 2010) and calculated in PyCO2SYS:

$$\gamma_{T_\text{C}} = \left(\frac{\partial \ln[\text{CO}_2(\text{aq})]}{\partial T_\text{C}}\right)_{A_\text{T}}^{-1} = \left(\frac{\partial T_\text{C}}{\partial \text{pH}}\right)_{A_\text{T}} \left(\frac{\partial \ln[\text{CO}_2(\text{aq})]}{\partial \text{pH}}\right)_{A_\text{T}}^{-1} \tag{E1}$$

$$\gamma_{A_\text{T}} = \left(\frac{\partial \ln[\text{CO}_2(\text{aq})]}{\partial A_\text{T}}\right)_{T_\text{C}}^{-1} = \left(\frac{\partial A_\text{T}}{\partial \text{pH}}\right)_{T_\text{C}} \left(\frac{\partial \ln[\text{CO}_2(\text{aq})]}{\partial \text{pH}}\right)_{T_\text{C}}^{-1} \tag{E2}$$

$$\beta_{T_\text{C}} = \left(\frac{\partial \ln[\text{H}^+]}{\partial T_\text{C}}\right)_{A_\text{T}}^{-1} = -\log_{10}(e) \left(\frac{\partial T_\text{C}}{\partial \text{pH}}\right)_{A_\text{T}} \tag{E3}$$

$$\beta_{A_\text{T}} = \left(\frac{\partial \ln[\text{H}^+]}{\partial A_\text{T}}\right)_{T_\text{C}}^{-1} = -\log_{10}(e) \left(\frac{\partial A_\text{T}}{\partial \text{pH}}\right)_{T_\text{C}} \tag{E4}$$

where $e$ is Euler's number (2.71828...).

For the saturation-state buffers $\omega_{T_C}$ and $\omega_{A_T}$ we also evaluate

- $(\partial \ln \Omega / \partial [CO_3^{2-}])$ by AD of the natural log of $\Omega(\text{aragonite})$, calculated with Eq. (D1), with respect to $[CO_3^{2-}]$ (note that this is the same value as for $\Omega(\text{calcite})$, due to the logarithm and the fact that these terms differ by the constant ratio of their solubility products);

- $(\partial [CO_3^{2-}] / \partial \text{pH})_{T_C}$ by AD of Eq. (C8) with respect to pH;

- $(\partial [CO_3^{2-}] / \partial \text{pH})_{A_T}$ by AD of Eq. (C8), substituting $T_C$ by Eq. (C9), with respect to pH.

The buffer factors are then given by

$$\omega_{T_C} = \left( \frac{\partial \ln \Omega}{\partial T_C} \right)_{A_T}^{-1} = \left( \frac{\partial T_C}{\partial \text{pH}} \right)_{A_T} \left( \frac{\partial \ln \Omega}{\partial [CO_3^{2-}]} \right)_{A_T}^{-1} \left( \frac{\partial [CO_3^{2-}]}{\partial \text{pH}} \right)_{A_T}^{-1} \tag{E5}$$

$$\omega_{A_T} = \left( \frac{\partial \ln \Omega}{\partial A_T} \right)_{T_C}^{-1} = \left( \frac{\partial A_T}{\partial \text{pH}} \right)_{T_C} \left( \frac{\partial \ln \Omega}{\partial [CO_3^{2-}]} \right)_{T_C}^{-1} \left( \frac{\partial [CO_3^{2-}]}{\partial \text{pH}} \right)_{T_C}^{-1} \tag{E6}$$

The approach taken here avoids AD evaluations over the iterative solvers, because while possible, that is computationally slower than over non-iterative functions.

### E2   Revelle factor

The Revelle factor ($R_F$; Broecker et al., 1979) is computed from $T_C$ and $\gamma_{T_C}$, with the latter evaluated as described in Sect. E1,

following Egleston et al. (2010):

$$R_F = \left( \frac{\partial f_{CO_2}}{\partial T_C} \right) \left( \frac{T_C}{f_{CO_2}} \right) = \frac{T_C}{\gamma_{T_C}} \tag{E7}$$

### E3   Isocapnic quotient and $\psi$

To evaluate the isocapnic quotient ($Q$) of Humphreys et al. (2018), we first evaluate the derivatives

- $(\partial T_C / \partial \text{pH})_{f_{CO_2}}$ by AD of Eq. (C14) with respect to pH;

- $(\partial A_T / \partial \text{pH})_{f_{CO_2}}$ by AD of Eq. (B1), using Eq. (B6) for the $A_C$ term, with respect to pH.

The isocapnic quotient is defined and calculated in PyCO2SYS as follows:

$$Q = \left( \frac{\partial A_T}{\partial T_C} \right)_{f_{CO_2}} = \left( \frac{\partial A_T}{\partial \text{pH}} \right)_{f_{CO_2}} \left( \frac{\partial T_C}{\partial \text{pH}} \right)_{f_{CO_2}}^{-1} \tag{E8}$$

Finally, the 'released $CO_2$:precipitated carbonate ratio' ($\psi$) of Frankignoulle et al. (1994) is calculated following Humphreys et al. (2018):

$$\psi = \frac{2}{Q} - 1 \tag{E9}$$

## Appendix F: Initial pH estimate when solving from $A_T$ and $T_C$

For clarity in the equations in this section, we abbreviate $[H^+]$ as $h$.

Following Munhoven (2013), carbonate-borate alkalinity ($A_{CB}$) from Eq. (3) as a function of $T_C$ and $h$ is

$$A_{CB}(h, T_C) = \frac{T_C K_1^* (h + 2 K_2^*)}{h^2 + K_1^* h + K_1^* K_2^*} + \frac{T_B K_B^*}{h + K_B^*} \tag{F1}$$

This can be rearranged into a third-order polynomial in $h$:

$$P_{T_C}(h) = h^3 + h^2 g_2(T_C) + h g_1(T_C) + g_0(T_C) = 0 \tag{F2}$$

where

$$g_2(T_C) = K_B^* \left( 1 - \frac{T_B}{A_{CB}} \right) - K_1^* \left( 1 - \frac{T_C}{A_{CB}} \right) \tag{F3}$$

$$g_1(T_C) = K_1^* \left[ K_B^* \left( 1 - \frac{T_B + T_C}{A_{CB}} \right) + K_2^* \left( 1 - \frac{2 T_C}{A_{CB}} \right) \right] \tag{F4}$$

$$g_0(T_C) = K_1^* K_2^* K_B^* \left( 1 - \frac{2 T_C + T_B}{A_{CB}} \right) \tag{F5}$$

The initial $h$ value is determined by

$$h_0(T_C) = \begin{cases} 10^{-3} & \text{for } A_T \leq 0 \\ h_{min} + \sqrt{-\frac{P_{T_C}(h_{min})}{\sqrt{g_2^2 - 3 g_1}}} & \text{for } A_T > 0 \\ 10^{-10} & \text{for } A_T \geq 2 T_C + T_B \end{cases} \tag{F6}$$

where $h_{min}$ is defined in Eq. (10). Negative $A_{CB}$ is impossible because its equation contains only positive terms, so the equations above cannot be applied if $A_T$ is indeed negative. The default $h_0$ of $10^{-3}$ mol·kg$^{-1}$, corresponding to a pH of 3, is therefore used for that case (e.g. after the alkalinity end-point in an acidimetric titration). The maximum possible $A_{CB}$ is $2 T_C + T_B$, where $T_C$ is entirely $CO_3^{2-}$ and $T_B$ is entirely $B(OH)_4^-$. Where $A_T$ is actually higher than this limit of this simplified expression, we expect a high pH (given the dominance of $CO_3^{2-}$ within $T_C$) and therefore use an initial estimate pH of 10. Otherwise, $h_{min}$ in Eq. (F6) is found using Eq. (10). A default $h_0$ of $10^{-7}$ mol·kg$^{-1}$ (pH 7) is used when $g_2^2 - 3 g_1 \leq 0$ in Eq. (F6) (Munhoven, 2013).

## Appendix G: Revelle factor calculation errors in older versions of CO2SYS-MATLAB

Older versions of CO2SYS-MATLAB, including v2.0.5 (Orr et al., 2018) from which PyCO2SYS was originally converted, have minor errors in how the Revelle factor is evaluated. These have been corrected in PyCO2SYS (also in CO2SYS-MATLAB v3.2.0 and CO2SYS-Excel v3), leading to small differences in the calculated values. These differences are on the order of 0.1; for context, the Revelle factor typically has a value on the order of 10. The differences are thus notable from a computational

perspective (i.e. many orders of magnitude greater than solver tolerance and floating point errors) but still mostly negligible in practical applications.

Rather than being corrected explicitly in PyCO2SYS, these errors are corrected automatically thanks to the approach of using automatic differentiation instead of finite-difference derivatives. The key errors in the original CO2SYS-MATLAB implementation of the finite-difference approach are

1. An incorrect reference $T_C$ value is used in the final evaluation. Rather than using the 'central' $T_C$ value, the change in $p_{CO_2}$ is divided by the adjusted $(T_C - \Delta T_C)$.

2. Under output conditions, the 'Peng correction' is not included in the evaluation of the Revelle factor (Sect. 2.2).

The lesser accuracy of the finite-difference method relative to automatic differentiation, particularly given the relatively large $\Delta T_C$ used in the original finite-difference implementation (i.e. $1 \, \mu\mathrm{mol} \cdot \mathrm{kg}^{-1}$), explains the differences between the two approaches that remains after the errors above have been corrected.

## Appendix H: Fixed $\Delta a$ values for uncertainty analysis

**Table H1.** Fixed $\Delta a$ values for uncertainty analysis.

| Argument(s) | $-\log_{10}\Delta a$ | Argument(s) | $-\log_{10}\Delta a$ |
|---|---|---|---|
| Core parameters | 4 | Pressure | 3 |
| $K_{\mathrm{NH_3}}^*$ | 16 | Salinity | 3 |
| $K_{\mathrm{sp}}^*$(aragonite) | 13 | Temperature | 3 |
| $K_{\mathrm{SO_4}}^*$ | 7 | $T_\alpha$ | 3 |
| $K_{\mathrm{B}}^*$ | 15 | $T_{\mathrm{NH_3}}$ | 3 |
| $K_{\mathrm{sp}}^*$(calcite) | 13 | $T_\beta$ | 3 |
| $K_1^*$ | 12 | $T_{\mathrm{B}}$ | 3 |
| $K_2^*$ | 15 | $T_{\mathrm{Ca}}$ | 3 |
| $K_{\mathrm{CO_2}}^*$ | 8 | $T_{\mathrm{F}}$ | 3 |
| $K_{\mathrm{HF}}^*$ | 9 | $T_{\mathrm{P}}$ | 3 |
| $K_{\mathrm{P1}}^*$ | 4 | $T_{\mathrm{Si}}$ | 3 |
| $K_{\mathrm{P2}}^*$ | 12 | $T_{\mathrm{SO_4}}$ | 3 |
| $K_{\mathrm{P3}}^*$ | 15 | $T_{\mathrm{H_2S}}$ | 3 |
| $K_{\mathrm{Si}}^*$ | 16 | $P_v$ | 5 |
| $K_{\mathrm{H_2S}}^*$ | 13 | $P_a$ | 4 |
| $K_w^*$ | 20 | $G$ | 5 |
| Any p$K^*$ | 4 | $R$ | 4 |

## Appendix I: Set-up for computational speed testing

The computational speed tests described in Sect. 5.4 were run on an HP Spectre x360 laptop with Intel Core i7-8565U CPU (1.80 GHz) and 16 GB of RAM. The operating system was Windows 10.

The Python tests were run using Python v3.9.7, Autograd v1.3, NumPy v1.21.2, and PyCO2SYS v1.8.0.

The MATLAB tests were run using MATLAB R2019b (Update 9) and CO2SYS-MATLAB v3.2.0.

The GNU Octave tests were run using GNU Octave v6.3.0 via its command-line interface and CO2SYS-MATLAB v3.2.0.

*Author contributions.* **MPH:** Conceptualisation, Methodology, Software, Validation, Writing—Original Draft, Visualisation. **ERL:** Software, Writing—Review & Editing. **JDS:** Software, Validation, Writing—Review & Editing. **DP:** Software, Writing—Review & Editing.

*Competing interests.* The authors declare that they have no competing interests.

*Acknowledgements.* We thank Doug Wallace for providing useful comments on this manuscript and we acknowledge his important role in the creation of the original CO2SYS software. We further acknowledge the developers of all subsequent versions of CO2SYS upon whose work PyCO2SYS was built. We thank Luke Gregor, Daniel Sandborn and Abigail Schiller for code contributions including extending the range of data types with which PyCO2SYS can be used. We are grateful to Guy Munhoven and James Orr for their detailed and constructive reviews.

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
