# Peer review of "PyCO2SYS v1.8: marine carbonate system calculations in Python"

_Geoscientific Model Development, 2021_

## Referee Comment (RC1)

**Review of**

**PyCO2SYS v1.7:** marine carbonate system calculations in Python**

submitted to *Geoscientific Model Development* by Matthew P. Humphreys et al.

**1** General comments**

Matthew P. Humphreys and co-authors present PyCO2SYS, a Python version of the "industry-standard" carbonate chemistry calculation package CO2SYS, originally developed for DOS by Lewis and Wallace (1998) and over the years ported to MATLAB and Microsoft Excel. The port to Python presented here derives from the MATLAB version 2.0.5 of CO2SYS (Orr et al., 2018), but includes extensions from the v3 branch of CO2SYS-MATLAB up to version 3.2 (Sharp et al., 2021). In addition to these improvements, PyCO2SYS also includes new developments not found in the MATLAB sub-family, such as automatic differentiation. The amendments required to use input data pairs that could not be processed with CO2SYS-MATLAB v2.0.5 (e.g.,  $[HCO_3^-]$  or  $[CO_3^{2-}]$  in combination with any other) are presented and a detailed assessment of the (generally negligible) differences between the results obtained with previous versions of CO2SYS is provided. What I find missing is an analysis of the computational performance and an exploration of the robustness of the adopted numerical procedures.

The paper fits well into the scope of *Geoscientific Model Development*. It is well written, goes to an informative level of detail and yet remains well readable. The material presentation is structured in an easy-to-follow way. There are a few minor imprecisions and inaccuracies here and there; a few things are missing in the analysis and discussion. These shortcomings are nevertheless relatively minor and can certainly be easily fixed. Until now, no similarly comprehensive carbonate chemistry calculation package has been available for Python. I am only aware of only two others, but that do not offer the same level of functionality: mocsy 2.0 and cbsyst 0.3.7. Making CO2SYS available for the ever growing Python users community is a highly welcome move and will certainly contribute to further increase its popularity and usefulness.

**2 Specific comment**

**Auto differentiation**

The automatic differentiation, performed with the help of the Python package Autopar is highlighted as one of the distinguishing features of PyCO2SYS compared to its predecessors.

It is, however, not explained to which parts of the calculations this automatic differentiation is applied, and also not to what level: is it only used to differentiate the expressions of the different terms of  $A_{\rm T}$ , or also for the parametrisations of the chemical constants, and if so, with respect to which variables?

If it is only applied to the terms of  $A_T$ , automatic differentiation might actually make things unnecessarily inefficient. The derivatives of the rational function terms that make up  $A_T$ , which are all that would be required for the Newton-Raphson solver, are actually quite straightforward and could just as well implemented manually once and for all, instead of having to rely upon a package that is adding another (unnecessary) layer of complexity and that is not actively developed any longer.

Finally, where automatic differentiation could really become useful, i.e., when it comes to uncertainty propagation, automatic differentiation has to be abandoned because it is computationally inefficient.

**3 Minor points and technical corrections**

**Abstract, Lines 15–16:**

"We discuss new insights that arose during the development process, for example that the marine carbonate system cannot be unambiguously solved from the total alkalinity and carbonate ion parameter pair."

The insights referred to in this sentence are actually not new, but simply littleknown. The topology of the  $CO_3^{2-}$  concentration isolines in  $T_C$ - $A_T$  space has been known for more than fifty years (Deffeyes, 1965). The existence of two pH roots for *most* pairs of  $A_T$ –[ $CO_3^{2-}$ ] values is an immediate consequence of that topology. Twenty years ago, Zeebe and Wolf-Gladrow (2001, pp. 276–277) also acknowledged the existence of two roots for this problem and recommend to use the larger one (in terms of [H+], i.e., the lower one in terms of pH). Please refer to Munhoven (2021) for a comprehensive discussion and a quantitative approach to characterise the  $A_C$ – $CO_3^{2-}$  problem (number of roots, bracketing intervals for individual roots and individual starting values).

It might also be worth mentioning that the  $T_{\rm C}$ –HCO3- problem is affected by similar ambiguities, acknowledged again shortly by Zeebe and Wolf-Gladrow (2001, pp. 276), and analysed and discussed in more detail by Munhoven (2021, Appendix A).

**Page 5, Table 2:** While I understand that salinity is zero for freshwater, I do not see why all the total concentrations from ammonia to sulfide have to be zero for freshwater. Had these values not better be left under the control of the user?

The table's footnote *b* states

"In GEOSECS-Peng, phosphate is not included in the definition of total alkalinity."

Something must be wrong here. Since the very first version of CO2SYS, option 7 (originally "Peng", now "GEOSECS-Peng") included the contribution of the

phosphates to total alkalinity, although in a peculiar way based upon a charge weighted approach, instead of the proton donor/proton acceptor approach of Dickson (1981). Option 6 (originally "GEOSECS", now "GEOSECS-Takahashi"), on the other hand, used the  $A_{CB}$  approximation to  $A_{T}$ . Please clarify.

The information in the GEOSECS column is anyway not entirely clear: neither GEOSECS-Takahashi nor GEOSECS-Peng include contributions from ammonia or sulfides to  $A_T$  – the meaning of "User-defined" for these two at least is not obvious. GEOSECS-Takahashi only considers carbonate and borate alkalinity in their  $A_T$  definition: so what does PyCO2SYS do with the extra user-defined data?

This should be presented more precisely.

**Page 11, lines 209:** The implementation in PyCO2SYS actually follows the approach of Munhoven (2013a) more or less exactly. The subroutine ahini\_for\_at in phsolvers.f90 from mocsy 2.0 was actually taken from SolveSAPHE 1.0.1 and the only adaptations made relate to the way the values for the chemical constants are passed (cf. subroutine AHINI\_FOR\_AT in mod\_phsolvers.f90 which is contained in the codes included in Munhoven (2013a, Supplement) or from Munhoven (2013b)).

**Page 11, lines 225–226:**

"[...], so the approach of Munhoven (2013a) cannot be applied if  $A_T$  is indeed negative (e.g. after the alkalinity end-point in an acidimetric titration)."

This statement is in clear contradiction with the SolveSAPHE code provided by Munhoven (2013a, Supplement) which shows that if  $A_T < 0$ ,  $h_0$  is set to  $10^{-3}$  mol/kg and if  $A_T > 2T_C + T_B$ ,  $h_0$  is set to  $10^{-10}$  mol/kg (see next comment). The approach of Munhoven (2013a) thus obviously also considers the case  $A_T < 0$  and addresses it exactly the same way as ... PyCO2SYS.

Please check out the original code and rewrite this sentence more accurately (see also previous comment).

**Page 12, line 230:** Although  $h_{\min}$  indeed always has a real value (and even has a real *positive* value), that value does not always lead to a meaningful  $h_0(s)$  as the resulting  $A_{CB}$  may possibly be greater than  $2T_C + T_B$ , which is not possible  $h_0(s)$  thus actually has to fulfil some conditions to keep  $A_{CB}$  within bounds, as explained in the "Mathematical and Technical Details" memo in the Supplement to Munhoven (2021).

Starting a Newton-Raphson iteration with physically meaningless initial values may jeopardize convergence, which must of course be avoided.

**Page 13, line 263:** Is the reference to Cai et al. (2017) for the ammonia and sulfide extensions of CO2SYS correct? I would rather have expected Xu et al. (2017).

**Page 14, line 287:** It would be good to cite the paper by Hagens and Middelburg (2016) here, which also deals with various aspects of the calculation of generalised buffer factors.

**Page 14, line 296–306:** Do I understand this correctly: with the 'automatic' approach, all the buffer factors are calculated analytically, in a way that is fully consistent with the currently adopted  $A_T$  composition; with the 'explicit' approach, all the buffer factors are calculated from equations taken from the literature and that may rely on simplified  $A_T$  composition, except for the Revelle factor, for which only a finite difference approximation is calculated, for compatibility reasons with previous CO2SYS versions?

It is somehow unsatisfactory that no direct equation was implemented for the Revelle factor, which remains the best known of all buffer factors at the end of the day. Why not add a third approach ('legacy' or similar), which would only provide the finite difference approximation for the Revelle factor, and then include a direct equation for the Revelle factor with the 'explicit' approach and so bring it on par with the others.

Finally: please provide us with individual references for each buffer factor equation actually implemented.

**Page 17, line 337–338:**

"We use finite differences rather than automatic differentiation here because the latter, while possible, is computationally inefficient to apply over the entire PyCO2SYS program."

Why is this so computationally inefficient? Is this due to the usage of the Autograd package? If only the derivatives of the alkalinity parts are required, why not implement them manually once and for all? These are comparatively straightforward. Now, the derivatives of the chemical constants might possibly be required as well—the text does unfortunately not include enough information about the level to which the differentiation is pushed. If so this approach would indeed be unrealistic. As requested in the specific comments above, additional information about which expressions exactly in the code automatic differentiation is applied to would be helpful.

Page 17, line 339: duplicate "an"

Page 18, line 385: duplicate "than"

**Page 19, line 396:** I suggest to rename section 4.2 to "Comparison with previous versions of CO2SYS" — "other software" is somewhat misleading as only CO2SYS variants are considered in this discussion.

**Page 24, line 496:** From experience, I think that "often" better had to read "most often" or "generally" in this context.

**Page 25, line 504:**

"Which root the solver finds depends on the initial pH estimate [...]"

This is a rather unsatisfactory behaviour as it makes it impossible to foresee whether the pH calculation can terminate reliably. Such unpredictable (and therefore unwanted) behaviour can nevertheless be safely avoided by first proceeding to a root localisation, and then using a bracketing root-finding algorithm instead of a plain Newton-Raphson (see, e.g., Munhoven, 2021).

**Page 24, line 552–554:**

"Now that the basis of PyCO2SYS is established, we would welcome more direct interaction with the groups developing these other tools, working towards a set of marine-carbonate-system-solving tools that return identical results regardless of the software platform."

This is an excellent idea. Such a joint effort could also define standard benchmark problems in order to assess the numerical and computational performances of the different tools.

**Page 28**, *Code availability* section: Does PyCO2SYS require any particular version of Python? If so, it would be good to state this here.

**Page 29, Eq. (B2):** The expression for  $A_w$  should actually read

$$A_{\rm w} = [{\rm OH}^-] - [{\rm H}^+]_{\rm free} = rac{K_{\rm w}^*}{[{\rm H}^+]} - rac{[{\rm H}^+]}{phcvt},$$

where *phcvt* is a factor to convert from the free to the working (total?) pH scale. This nevertheless appears to be correctly implemented in the code.

**Page 32, Eq. (B20):** This is not in line with the PyCO2SYS code. Since  $K_{SO4}^*$  is on the free pH scale, [H+] must actually be [H+]free in this equation.

**Page 32, Eq. (B22):** This is not in line with the PyCO2SYS code. Since  $K_F^*$  is on the free pH scale, [H+] must actually be [H+]free in this equation

**Page 32, Eq. (B24):** This extension is not consistent with the definition of total alkalinity (Dickson, 1981), which states that

"[t]he total alkalinity of a natural water is thus defined as the number of moles of hydrogen ion equivalent to the excess of proton acceptors (bases formed from weak acids with a dissociation constant  $K \leq 10^{-4.5}$ , at 25 °C and zero ionic strength) over proton donors (acids with  $K > 10^{-4.5}$ ) in one kilogram of sample." So, it is not  $K_{\alpha}^*$  that has to be compared against this decisive threshold, but  $K_{\alpha}$ , and not at any arbitrary combination of temperature, salinity and pressure, but at a clearly defined set. How reliable is the adopted approach, given that approximation?

**Page 32, line 686:** According to Wolf-Gladrow et al. (2007) the zero level of protons is a *species*, not a concentration or some other pZLP. Each acid-base system actually has a different zero level of protons. In order to define the zero levels of protons for a complex mixture of acids and bases in a consistent way, Wolf-Gladrow et al. (2007) call upon a threshold pK value (their Sect. 2.4.4). They denote that threshold pK value by  $pK_{zlp}$  (it would be recommendable to stick to that notation in order to avoid unnecessary confusion). The currently accepted definition of  $A_T$  by Dickson (1981) is thus based upon  $pK_{zlp} = 4.5$ . However,  $pK_{zlp}$  is not the zero level of protons (there as many zero levels of protons as there are acid-base systems in the solution). Please rewrite this more precisely.

**Page 34, Sect. C2.2:** It might be worth mentioning that some pairs of  $A_T$ –pH data values lead to negative  $A_C$  (see, e.g., Munhoven, 2021, for illustrative examples), which is physically impossible. Does pyCO2SYS catch that kind of exception?

**Page 34, Sect. C2.9:** Similarly to the  $A_T$ –CO32– problem, the  $T_C$ –HCO3– problem may have zero, one or two pH roots. There is actually an upper limit, which is strictly lower than 1, for the [HCO3–]/ $T_C$  fraction above which there is no solution; at that exact limit, there is one, and below that limit there are two roots. Zeebe and Wolf-Gladrow (2001, p. 276) already mention the possibility of two roots and recommend to chose the low-[H+] (high-pH) one, without any further explanation though. Munhoven (2021, Appendix A) provides a comprehensive analysis of this problem and a short discussion about the respective side effects of each one of the two roots, which may contribute to discriminate between the two and help to chose the relevant one.

**Page 37, paragraph at lines 802–808:** There are unfortunately several inaccurate statements in this paragraph. The described procedure indeed follows Orr and Epitalon (2015) who follow ... Munhoven (2013a). The solver engines in mocsy 2.0 (Orr and Epitalon, 2015) stem from SolveSAPHE 1.0.1 (Munhoven, 2013a, Supplement) – this is clearly stated in the comments in the mocsy 2.0 code. The initialisation subroutine from SolveSAPHE — which was included in mocsy 2.0 modified only to transfer the chemical constants differently — furthermore uses a fall-back value  $h_0(T_{\rm C}) = 10^{-7}$  mol/kg in case  $g_2^2 - 3g_1 \leq 0$ , not mentioned in the text here, but nevertheless implemented exactly that same way in PyCO2SYS (according to solve/initialise.py).

Please have a look at the *original* code (also available from Munhoven, 2013b) and rewrite this paragraph more accurately.

**Page 45, line 1039:** The DOI provided for CO2SYS-MATLAB v1.1 has been dead for a long time. It does not resolve correctly since the CDIAC collections were moved to ESS-DIVE. Perhaps, you may get this problem fixed upstream, otherwise, it would be best to remove that DOI.

**References**

- K. S. Deffeyes. Carbonate equilibria : A graphic and algebraic approach. *Limnol. Oceanogr.*, 10(3):412–426, 1965. doi: 10.4319/lo.1965.10.3.0412.
- A. G. Dickson. An exact definition of total alkalinity and a procedure for the estimation of alkalinity and total inorganic carbon from titration data. *Deep-Sea Res. A*, 28(6):609–623, 1981. doi: 10.1016/0198-0149(81)90121-7.
- M. Hagens and J. J. Middelburg. Generalised expressions for the response of pH to changes in ocean chemistry. *Geochim. Cosmochim. Ac.*, 187:334–349, 2016. doi: 10.1016/j.gca.2016.04.012.
- E. Lewis and D. Wallace. Program developed for CO2 system calculations. Technical Report 105, Carbon Dioxide Analysis Center, Oak Ridge National Laboratory, Oak Ridge (TN), 1998. URL http://cdiac.ornl.gov/oceans/ co2rprt.html.
- G. Munhoven. Mathematics of the total alkalinity-pH equation pathway to robust and universal solution algorithms: the SolveSAPHE package v1.0.1. *Geosci. Model Dev.*, 6(4):1367–1388, 2013a. doi: 10.5194/gmd-6-1367-2013.
- G. Munhoven. SolveSAPHE (Solver Suite for Alkalinity-PH Equations), v1.0.1, August 2013b.
- G. Munhoven. SolveSAPHE-r2 (v2.0.1): revisiting and extending the Solver Suite for Alkalinity-PH Equations for usage with CO2, HCO3- or CO32- input data. *Geosci. Model Dev.*, 14(7):4225–4240, 2021. doi: 10.5194/gmd-14-4225-2021.
- J. C. Orr and J.-M. Epitalon. Improved routines to model the ocean carbonate system : mocsy 2.0. *Geosci. Model Dev.*, 8(3):485–499, 2015. doi: 10.5194/gmd-8-485-2015.
- J. C. Orr, J.-M. Epitalon, A. G. Dickson, and J.-P. Gattuso. Routine uncertainty propagation for the marine carbon dioxide system. *Mar. Chem.*, 207:84–107, 2018. doi: 10.1016/j.marchem.2018.10.006.
- J. D. Sharp, D. Pierrot, M. P. Humphreys, J.-M. Epitalon, J. C. Orr, E. R. Lewis, and D. W. R. Wallace. CO2SYSv3 for MATLAB, v3.2, May 2021.
- D. A. Wolf-Gladrow, R. E. Zeebe, C. Klaas, A. Körtzinger, and A. G. Dickson. Total alkalinity : The explicit conservative expression and its application to biogeochemical processes. *Mar. Chem.*, 106(1-2):287–300, 2007. doi: 10.1016/j. marchem.2007.01.006.

- Y.-Y. Xu, D. Pierrot, and W.-J. Cai. Ocean carbonate system computation for anoxic waters using an updated CO2SYS program. *Mar. Chem.*, 195:90–93, 2017. doi: 10.1016/j.marchem.2017.07.002.
- R. E. Zeebe and D. Wolf-Gladrow. CO2 in seawater : Equilibrium, kinetics, isotopes, volume 65 of Elsevier Oceanography Series. Elsevier, Amsterdam (NL), 2001. ISBN 978-0-444-50579-8. URL http://www.sciencedirect.com/ science/bookseries/04229894/65.

Liège, 8th July 2021 Guy Munhoven

---

## Author Comment (AC1)

Normal text: comment by reviewer
*Bold italic black: response by us*

**Reviewer 1 (Munhoven)**

**General comments**

[M1] Matthew P. Humphreys and co-authors present PyCO2SYS, a Python version of the "industry-standard" carbonate chemistry calculation package CO2SYS, originally developed for DOS by Lewis and Wallace (1998) and over the years ported to MATLAB and Microsoft Excel. The port to Python presented here derives from the MATLAB version 2.0.5 of CO2SYS (Orr et al., 2018), but includes extensions from the v3 branch of CO2SYS-MATLAB up to version 3.2 (Sharp et al., 2021). In addition to these improvements, PyCO2SYS also includes new developments not found in the MATLAB sub-family, such as automatic differentiation. The amendments required to use input data pairs that could not be processed with CO2SYS-MATLAB v2.0.5 (e.g., [HCO$_{-3}$] or [CO$_{2-3}$] in combination with any other) are presented and a detailed assessment of the (generally negligible) differences between the results obtained with previous versions of CO2SYS is provided. What I find missing is an analysis of the computational performance and an exploration of the robustness of the adopted numerical procedures.

The paper fits well into the scope of Geoscientific Model Development. It is well written, goes to an informative level of detail and yet remains well readable. The material presentation is structured in an easy-to-follow way. There are a few minor imprecisions and inaccuracies here and there; a few things are missing in the analysis and discussion. These shortcomings are nevertheless relatively minor and can certainly be easily fixed. Until now, no similarly comprehensive carbonate chemistry calculation package has been available for Python. I am only aware of only two others, but that do not offer the same level of functionality: mocsy 2.0 and cbsyst 0.3.7. Making CO2SYS available for the ever growing Python users community is a highly welcome move and will certainly contribute to further increase its popularity and usefulness.

*We thank the reviewer for their positive assessment of our manuscript and for the many useful comments and suggestions that follow.*

*We have added a new section in the Discussion (5.4) that compares the computational speed of PyCO2SYS with CO2SYS-MATLAB.*

**Specific comments**

Auto differentiation

[M2] The automatic differentiation, performed with the help of the Python package Autopar is highlighted as one of the distinguishing features of PyCO2SYS compared to its predecessors.

It is, however, not explained to which parts of the calculations this automatic differentiation is applied, and also not to what level: is it only used to differentiate the expressions of the different terms of $A_T$, or also for the parametrisations of the chemical constants, and if so, with respect to which variables?

*We have revised Section 3.1.1 (Automatic differentiation) to more clearly state what is evaluated using this technique (i.e. [1] the derivatives of the main chemical speciation function needed for the alkalinity-pH solvers and [2] chemical buffer factors).*

*We also revised Sections C2.1, C2.3, C2.4 and C2.5 in Appendix C (Solving routines) to explicitly describe where AD is used in the solver process.*

*We also added a new Appendix to define all the buffers computed by PyCO2SYS and to show exactly how AD is used in each case.*

[M3] If it is only applied to the terms of $A_T$, automatic differentiation might actually make things unnecessarily inefficient. The derivatives of the rational function terms that make up $A_T$, which are all that would be required for the Newton-Raphson solver, are actually quite straightforward and could just as well implemented manually once and for all, instead of having to rely upon a package that is adding another (unnecessary) layer of complexity and that is not actively developed any longer.

*Automatic differentiation is not only used for the Newton-Raphson solvers but also for calculating all buffer factors and related properties (e.g. isocapnic quotient). Our Autograd-based approach means that all the solvers and all these additional properties will still be correctly*

*computed following any future changes in the main chemical speciation function. The manual implementation requires more work, and has more opportunities for errors, than implied by the reviewer's comment. Autograd may add a layer of behind-the-scenes complexity in terms of managing your Python installation, but it reduces complexity in the code of PyCO2SYS itself, and makes no difference in complexity for day-to-day use of PyCO2SYS.*

*Also, a manual implementation can never be 'once and for all' because there is always the possibility to add additional components or equilibrium modelling approaches given the chemical complexity of seawater. For example, earlier versions of CO2SYS did not include sulfide and ammonia equilibria, but their effects are now accounted for. While their effects are typically small in the open ocean and can probably often be ignored in the manually implemented equations, our approach will also still produce accurate results in unusual solution compositions that deviate far from this norm, which a user may wish to investigate.*

*We agree that the reliance on the Autograd package is less than ideal, but as we discussed in the manuscript (Section 5.4 - Outlook) the closest alternative, which is in very active development, does not yet work on Windows. Please note that we already maintain an independent fork of the Autograd package and it is this version that gets installed along with PyCO2SYS, so we are not reliant on other maintainers to make sure PyCO2SYS keeps working.*

[M4] Finally, where automatic differentiation could really become useful, i.e., when it comes to uncertainty propagation, automatic differentiation has to be abandoned because it is computationally inefficient.

*First, we disagree that this is the only application where automatic differentiation is 'really useful' (see answer to M3 above).*

*Second, automatic differentiation can be used for uncertainty propagation - it does not 'have' to be abandoned - if exact gradients are sought, it's just simpler and quicker not to use it for this particular application. Again, as noted in our response to the previous point [M3], we anticipate the speed will increase in the future as new software becomes more widely available.*

**Minor points and technical corrections**

[M5] Abstract, Lines 15–16:

"We discuss new insights that arose during the development process, for example that the marine carbonate system cannot be unambiguously solved from the total alkalinity and carbonate ion parameter Pair."

The insights referred to in this sentence are actually not new, but simply little-known. The topology of the $CO_3^{2-}$ concentration isolines in $T_C$-$A_T$ space has been known for more than fifty years (Deffeyes, 1965). The existence of two pH roots for most pairs of $A_T$–[$CO_3^{2-}$] values is an immediate consequence of that topology. Twenty years ago, Zeebe and Wolf-Gladrow (2001, pp. 276–277) also acknowledged the existence of two roots for this problem and recommend to use the larger one (in terms of [$H^+$], i.e., the lower one in terms of pH). Please refer to Munhoven (2021) for a comprehensive discussion and a quantitative approach to characterise the $A_C$–$CO_3^{2-}$ problem (number of roots, bracketing intervals for individual roots and individual starting values).

***Thank you for pointing this out, we have addressed this oversight as follows.***

***In the Abstract, we changed "We discuss new insights that arose during the development process, …" to "We discuss insights that guided the development, …"***

***We renamed Section 5.2 (Total alkalinity-carbonate ion parameter pair) to "Parameter pairs with multiple solutions" and adjusted the wording so we do not imply this is a novel finding. The focus is shifted to the conceptual analysis of the non-standard root (Figure 6 and associated text), which is novel to the best of our knowledge. We added citations of the studies mentioned by the reviewer in this section too.***

[M6] It might also be worth mentioning that the $T_C$–$HCO_3^-$ problem is affected by similar ambiguities, acknowledged again shortly by Zeebe and Wolf-Gladrow (2001, pp. 276), and analysed and discussed in more detail by Munhoven (2021, Appendix A).

***Thank you for pointing this out. We added a new section (5.2.2) to discuss this double root similar to that for the alkalinity-CO3 pair. See also [M31].***

[M7] Page 5, Table 2: While I understand that salinity is zero for freshwater, I do not see why all the total concentrations from ammonia to sulfide have to be zero for freshwater. Had these values not better be left under the control of the user?

***This behaviour is inherited from earlier versions of CO2SYS, which set phosphate and silicate to zero for the freshwater K1/K2 parameterisation. The main problem with allowing user-defined concentrations would be that the parameterisations of the equilibrium constants used for many of the relevant reactions are not applicable to freshwater; indeed, in Table 2, many of the relevant constants are undefined in the Freshwater case.***

***We agree this would be a nice addition to future versions but it may require detailed studies on each set of constants to create parameterisations that work across the full salinity range. We therefore find that it goes beyond the scope of this initial release.***

[M8] The table's footnote b states

> "In GEOSECS-Peng, phosphate is not included in the definition of total alkalinity."

> Something must be wrong here. Since the very first version of CO2SYS, option 7 (originally "Peng", now "GEOSECS-Peng") included the contribution of the phosphates to total alkalinity, although in a peculiar way based upon a charge weighted approach, instead of the proton donor/proton acceptor approach of Dickson (1981). Option 6 (originally "GEOSECS", now "GEOSECS-Takahashi"), on the other hand, used the $A_{CB}$ approximation to $A_T$. Please clarify.

***Changed to "In GEOSECS-Takahashi, phosphate is not included in the definition of total alkalinity; in GEOSECS-Peng, phosphate is included, though the contribution of each species to alkalinity is determined incorrectly, based on charge rather than a zero-level of protons at pK 4.5."***

[M9] The information in the GEOSECS column is anyway not entirely clear: neither GEOSECS-Takahashi nor GEOSECS-Peng include contributions from ammonia or sulfides to $A_T$ – the meaning of "User-defined" for these two at

least is not obvious. GEOSECS-Takahashi only considers carbonate and borate alkalinity in their $A_T$ definition: so what does PyCO2SYS do with the extra user-defined data?

This should be presented more precisely.

***Good point. We have now explained this subtlety in the opening to Section 3.3.1 (Additional alkalinity components).***

[M10] Page 11, lines 209: The implementation in PyCO2SYS actually follows the approach of Munhoven (2013a) more or less exactly. The subroutine `ahini_for_at` in `phsolvers.f90` from mocsy 2.0 was actually taken from SolveSAPHE 1.0.1 and the only adaptations made relate to the way the values for the chemical constants are passed (cf. subroutine `AHINI_FOR_AT` in `mod_phsolvers.f90` which is contained in the codes included in Munhoven (2013a, Supplement) or from Munhoven (2013b)).

***Updated to "Following Munhoven (2013) and as also implemented elsewhere (e.g. Orr et al.), …" to make it clearer that Munhoven (2013) is the original source.***

[M11] Page 11, lines 225–226:

"[. . . ], so the approach of Munhoven (2013a) cannot be applied if $A_T$ is indeed negative (e.g. after the alkalinity end-point in an acidimetric titration)." This statement is in clear contradiction with the SolveSAPHE code provided by Munhoven (2013a, Supplement) which shows that if $A_T < 0$, $h_0$ is set to $10^{-3}$ mol/kg and if $A_T > 2T_C + T_B$, $h_0$ is set to $10^{-10}$ mol/kg (see next comment). The approach of Munhoven (2013a) thus obviously also considers the case $A_T < 0$ and addresses it exactly the same way as . . . PyCO2SYS.

Please check out the original code and rewrite this sentence more accurately (see also previous comment).

***The problem here is that we were (wrongly) not considering the approach taken outside the cutoffs (i.e. AT < 0 or AT > 2TC + TB) to be part of "the approach of Munhoven (2013a)". The reviewer is quite right that this is indeed an integral part of the Munhoven (2013a) approach and so we have updated the wording accordingly. See also reviewer comment [M32].***

[M12] Page 12, line 230: Although $h_{min}$ indeed always has a real value (and even has a real positive value), that value does not always lead to a meaningful $h_0$ (s) as the resulting $A_{CB}$ may possibly be greater than $2T_C + T_B$, which is not possible $h_0$ (s) thus actually has to fulfil some conditions to keep $A_{CB}$ within bounds, as explained in the "Mathematical and Technical Details" memo in the Supplement to Munhoven (2021).

Starting a Newton-Raphson iteration with physically meaningless initial values may jeopardize convergence, which must of course be avoided.
***Thanks for pointing this out, we have added the additional constraint in the code and mentioned it here in the manuscript.***

[M13] Page 13, line 263: Is the reference to Cai et al. (2017) for the ammonia and sulfide extensions of CO2SYS correct? I would rather have expected Xu et al. (2017).
***Cai et al. describes the extensions while Xu et al. is the citation for the program itself. We have updated to cite both publications here.***

[M14] Page 14, line 287: It would be good to cite the paper by Hagens and Middelburg (2016) here, which also deals with various aspects of the calculation of generalised buffer factors.
***We added the suggested citation here.***

[M15] Page 14, line 296–306: Do I understand this correctly: with the 'automatic' approach, all the buffer factors are calculated analytically, in a way that is fully consistent with the currently adopted $A_T$ composition; with the 'explicit' approach, all the buffer factors are calculated from equations taken from the literature and that may rely on simplified $A_T$ composition, except for the Revelle factor, for which only a finite difference approximation is calculated, for compatibility reasons with previous CO2SYS versions?

It is somehow unsatisfactory that no direct equation was implemented for the Revelle factor, which remains the best known of all buffer factors at the end of the day. Why not add a third approach ('legacy' or similar), which would only provide the finite difference approximation for the Revelle factor, and then include a direct equation for the Revelle factor with the 'explicit' approach and so bring it on par with the others.

*See also the other reviewer's comment [O14]. The automatic approach uses automatic differentiation to calculate the buffers as described in the update manuscript. This is the default behaviour for PyCO2SYS; all other approaches to calculating these variables are of secondary importance in this context. We do not see the value of adding a third, again slightly different way of calculating the Revelle factor. If a user would need to obtain this, they could do so by taking the 'explicit' calculation of gamma_TC and using equation (E7) from the revised manuscript. We added a note to this effect to the end of section 3.3.4.*

[M16] Finally: please provide us with individual references for each buffer factor equation actually implemented.
*This should now be addressed by the new Appendix on the buffer factor calculations. See also our response to your point [M2].*

[M17] Page 17, line 337–338:
"We use finite differences rather than automatic differentiation here because the latter, while possible, is computationally inefficient to apply over the entire PyCO2SYS program."
Why is this so computationally inefficient? Is this due to the usage of the Autograd package? If only the derivatives of the alkalinity parts are required, why not implement them manually once and for all? These are comparatively straightforward. Now, the derivatives of the chemical constants might possibly be required as well—the text does unfortunately not include enough information about the level to which the differentiation is pushed. If so this approach would indeed be unrealistic. As requested in the specific comments above, additional information about which expressions exactly in the code automatic differentiation is applied to would be helpful.
*Regarding where AD is used: see our responses to your other related comments on how we have addressed this.*

*Regarding efficiency: it just happens that Autograd is not particularly efficient from a computational perspective, with some derivatives taking orders of magnitude longer to compute than the original functions, depending on the complexity of the code. Other implementations of automatic differentiation in Python are faster, as discussed in Section 5.5 of the revised manuscript, but for various*

*reasons are not yet suitable to use with PyCO2SYS (e.g. compatibility across operating systems). In the future we are interested to improve this aspect and indeed some preliminary work has been done. But PyCO2SYS as it currently is, is already a valuable tool, with a little slower but broadly comparable performance to the MATLAB (see new Section 5.4), and already perfectly usable for the same range of oceanographic applications.*

*See also reviewer comments [M15] and [O18].*

[M18] Page 17, line 339: duplicate "an"
*Fixed.*

[M19] Page 18, line 385: duplicate "than"
*Fixed.*

[M20] Page 19, line 396: I suggest to rename section 4.2 to "Comparison with previous versions of CO2SYS" — "other software" is somewhat misleading as only CO2SYS variants are considered in this discussion.
*Renamed to "Comparison with other CO2SYS software" (not all the other versions are 'previous').*

[M21] Page 24, line 496: From experience, I think that "often" better had to read "most often" or "generally" in this context.
*Changed to 'generally'.*

[M22] Page 25, line 504:
"Which root the solver finds depends on the initial pH estimate [. . . ]"
This is a rather unsatisfactory behaviour as it makes it impossible to foresee whether the pH calculation can terminate reliably. Such unpredictable (and therefore unwanted) behaviour can nevertheless be safely avoided by first proceeding to a root localisation, and then using a bracketing root-finding algorithm instead of a plain Newton-Raphson (see, e.g., Munhoven, 2021).
*We agree - our statement is a general comment on the Newton-Raphson method, not something specific to our program. Indeed, the paragraph following this sentence goes on to explain that we already first use a root localisation to avoid unpredictable behaviour, as the reviewer*

*suggests. We do not find it necessary to add further complexity to the existing solver by using a bracketing approach - CO2SYS has long filled most needs of the oceanographic community with plain Newton-Raphson and no root localisation.*

[M23] Page 24, line 552–554:
        "Now that the basis of PyCO2SYS is established, we would welcome more direct interaction with the groups developing these other tools, working towards a set of marine-carbonate-system-solving tools that return identical results regardless of the software platform."
        This is an excellent idea. Such a joint effort could also define standard benchmark problems in order to assess the numerical and computational performances of the different tools.
*We agree - thank you!*

[M24] Page 28, Code availability section: Does PyCO2SYS require any particular version of Python? If so, it would be good to state this here.
*We added this information to the end of the opening part of Section 4 (Validation).*

[M25] Page 29, Eq. (B2): The expression for $A_w$ should actually read
        [corrected equation]
where phcvt is a factor to convert from the free to the working (total?) pH scale.
        This nevertheless appears to be correctly implemented in the code.
*All the equations here assume [H+] is provided on the same scale as the equilibrium constants in each expression, and this is how the code is implemented. We consider the relevant pH scale conversion to be a separate step before using the equations in Appendix B (see Appendix A); it is not an intrinsic part of them. See also our response to [M26] below.*

[M26] Page 32, Eq. (B20): This is not in line with the PyCO2SYS code. Since $K_{*SO4}$ is on the free pH scale, [H+] must actually be [H+]$_{free}$ in this equation.
*We do not find this necessary. In all the equations throughout this Appendix, [H+] must be on the same pH scale as the equilibrium*

*constants being used for the calculation. We have added a note to this effect to the beginning of Appendix B. The fact that, in PyCO2SYS, the KSO4 constant is always on the Free scale, does not need to be re-emphasised in this equation, which is fundamentally no different from all the other systems. In principle, the KSO4 constant could be represented on a different pH scale; it just isn't in PyCO2SYS.*

[M27] Page 32, Eq. (B22): This is not in line with the PyCO2SYS code. Since $K_{*F}$ is on the free pH scale, [H+] must actually be [H+]$_{free}$ in this equation
*See response to [M26] above.*

[M28] Page 32, Eq. (B24): This extension is not consistent with the definition of total alkalinity (Dickson, 1981), which states that "[t]he total alkalinity of a natural water is thus defined as the number of moles of hydrogen ion equivalent to the excess of proton acceptors (bases formed from weak acids with a dissociation constant K ≤10$_{-4.5}$, at 25 ∘C and zero ionic strength) over proton donors (acids with K >10$_{-4.5}$) in one kilogram of sample."

So, it is not $K_{*\alpha}$ that has to be compared against this decisive threshold, but $K_\alpha$, and not at any arbitrary combination of temperature, salinity and pressure, but at a clearly defined set. How reliable is the adopted approach, given that approximation?

*Thanks for pointing this out. We separate arbitrary components into proton acceptors and donors based on the user-defined K values at the given conditions mostly for lack of a better method. Of course, this won't matter for pK values far from 4.5, as changes to T,S,P won't cause the pK to cross that threshold, but things become ambiguous for values near 4.5.*

*As such, we've added: "Though the definition of alkalinity (Dickson, 1981) states that species are separated into proton acceptors and donors based on their dissociation constant at zero ionic strength and 25 C, we use the user-defined dissociation constants at the given conditions because one cannot convert arbitrary dissociation constants to their alkalinity-relevant values. Interpretations of results when arbitrary components are supplied to PyCO2SYS with pK\* values close to 4.5 should consider this nuance."*

[M29] Page 32, line 686: According to Wolf-Gladrow et al. (2007) the zero level of protons is a species, not a concentration or some other pZLP. Each acid-base system actually has a different zero level of protons. In order to define the zero levels of protons for a complex mixture of acids and bases in a consistent way, Wolf-Gladrow et al. (2007) call upon a threshold pK value (their Sect. 2.4.4). They denote that threshold pK value by $pK_{zlp}$ (it would be recommendable to stick to that notation in order to avoid unnecessary confusion). The currently accepted definition of $A_T$ by Dickson (1981) is thus based upon $pK_{zlp}$ = 4.5. However, $pK_{zlp}$ is not the zero level of protons (there as many zero levels of protons as there are acid-base systems in the solution). Please rewrite this more precisely.
***We've changed this sentence to read: "...values, with a zero-level of protons corresponding to a pK of 4.5 (Dickson, 1981; Wolf-Gladrow et al., 2007)."***

[M30] Page 34, Sect. C2.2: It might be worth mentioning that some pairs of $A_T$–pH data values lead to negative $A_C$ (see, e.g., Munhoven, 2021, for illustrative examples), which is physically impossible. Does pyCO2SYS catch that kind of exception?
***Yes, this exception is caught (also similarly if fCO2, HCO3 or CO3 are impossibly high when any of these is paired with known DIC) and a NaN (not-a-number) is returned instead of computing a negative DIC. We added the suggested note.***

[M31] Page 34, Sect. C2.9: Similarly to the $A_T$–$CO_2^{-3}$ problem, the $T_C$–$HCO_3^-$ problem may have zero, one or two pH roots. There is actually an upper limit, which is strictly lower than 1, for the $[HCO_3^-]$/$T_C$ fraction above which there is no solution; at that exact limit, there is one, and below that limit there are two roots.
    Zeebe and Wolf-Gladrow (2001, p. 276) already mention the possibility of two roots and recommend to chose the low-$[H^+]$ (high-pH) one, without any further explanation though. Munhoven (2021, Appendix A) provides a comprehensive analysis of this problem and a short discussion about the respective side effects of each one of the two roots, which may contribute to discriminate between the two and help to chose the relevant one.
***See our response to [M6]. Further thanks for sharing this insight.***

[M32] Page 37, paragraph at lines 802–808: There are unfortunately several inaccurate statements in this paragraph. The described procedure indeed follows Orr and Epitalon (2015) who follow . . . Munhoven (2013a). The solver engines in mocsy 2.0 (Orr and Epitalon, 2015) stem from SolveSAPHE 1.0.1 (Munhoven, 2013a, Supplement) – this is clearly stated in the comments in the mocsy 2.0 code. The initialisation subroutine from SolveSAPHE — which was included in mocsy 2.0 modified only to transfer the chemical constants differently — furthermore uses a fall-back value $h_0 (T_C ) = 10^{-7}$ mol/kg in case $g_{22} - 3g_1 \leq 0$, not mentioned in the text here, but nevertheless implemented exactly that same way in PyCO2SYS (according to `solve/initialise.py`).

 Please have a look at the original code (also available from Munhoven, 2013b) and rewrite this paragraph more accurately.
***This is essentially the same comment as [M11] (but referring to a different part of our manuscript that almost repeats some earlier text). We have resolved it in a similar way and also noted the additional pH=7 fall-back.***

[M33] Page 45, line 1039: The DOI provided for CO2SYS-MATLAB v1.1 has been dead for a long time. It does not resolve correctly since the CDIAC collections were moved to ESS-DIVE. Perhaps, you may get this problem fixed upstream, otherwise, it would be best to remove that DOI.
***We have made enquiries to get this fixed (months ago - before submitting the first draft of this manuscript) but as it still does not appear to be resolved we have removed the DOI as suggested.***

**Reviewer 2 (Orr)**

**GENERAL COMMENTS**

[O1] Humphreys et al. have provided an excellent manuscript on a new package (PyCO2SYS), entirely written in Python, that compute marine CO2 system variables, buffer factors, and uncertainties. Although similar packages exist in other languages, this is the first time such a wide-based package has been available in Python itself. The package is based on CO2SYS-MATLAB,

hence its name, but is not just a simple translation, containing many novelties and the most recent updates. Furthermore, the coauthors include the primary expert that developed the original version of CO2SYS (for MS-DOS), the expert that developed the subsequent Excel and MATLAB versions, and another who added major new features to the MATLAB version. This wide expertise combined with the first authors' strong lead in adding novel features in Python lends great confidence to this initiative. PyCO2SYS will be a welcome addition to the community, especially for young scientists, many of whom are adopting Python as a primary coding language

The manuscript is generally well written and requires only minor revisions. My strongest complaint is that the section on uncertainty propagation contains an equation that is incorrect and that the related Appendix G should be deleted because it too strongly resembles previous work, both rather easy fixes. I would also like to see an evaluation of the computational efficiency of PyCO2SYS along the lines suggested by the other reviewer, Guy Munhoven. My minor comments are listed below.

***Thank you for the positive comments and detailed feedback and suggestions, which have improved the manuscript.***

***We have added a new section in the Discussion (5.4) that compares the computational speed of PyCO2SYS with CO2SYS-MATLAB.***

**ABSTRACT**

[O2] line 8: The authors state "but no fully functional and rigorously validated tool was previously available for Python". They might want to say "written in Python" since seacarb is available for use in Python (through Rmagic or rpy2).

***Updated following reviewer's suggestion.***

[O3] line 12: "every solute is included"? "every" is a dangerous word.

***Updated to "every modelled solute".***

**INTRODUCTION**

[O4] line 32: what the authors refer to as "metadata" is not. Metadata is data that provides information about other data.

***Updated 'metadata' to 'auxiliary data'.***

[O5] line 64: It would be useful for readers if the authors could mention that "molinity" (number of moles of solute per kilogram of the mixture). I think molinity is more specific than the definition of "substance content" that is given in the IUPAC (1997) Goldbook, where the latter is defined as "Amount-of-substance of a component divided by the mass of the system." Furthermore, the authors' use of "content" as shorthand for "substance content" may escape all readers that have not read the Introduction.

***We are not sure how it is possible to be much more specific than the definition of substance content quoted by the reviewer. 'Amount of substance' refers to moles (from the Gold Book, it is "a measure of the number of specified elementary entities. An elementary entity may be an atom, a molecule, an ion, an electron, any other particle or specified group of particles.") and 'mass of the system' is self-explanatory (system = solvent + solutes). On the other hand, as far as we're aware, 'molinity' is not formally defined by any recognised authority. Nevertheless, we have added a note here that 'substance content' is sometimes referred to as molinity.***

***We note that this issue was the subject of significant discussion among many authors during recent preparation of a manuscript on best practices in chemical oceanographic data reporting, and the approach we have taken here is consistent with those recommendations (Jiang et al., in review).***

*Li-Qing Jiang, Denis Pierrot, Rik Wanninkhof, Richard A. Feely, Bronte Tilbrook, Simone R Alin, Leticia Barbero, Robert H. Byrne, Brendan R. Carter, Andrew G Dickson, Jean-Pierre Gattuso, Dana Greeley, Mario Hoppema, Matthew P. Humphreys, Johannes Karstensen, Nico Lange, Siv K. Lauvset, Ernie Lewis, Are Olsen, Fiz F. Pérez, Christopher Sabine, Jonathan Sharp, Toste Tanhua, Tom Trull, Antón Velo, Andrew J. Allegra, Paul Barker, Eugene Burger, Wei-Jun Cai, Chen-Tung Arthur Chen, Jessica N Cross, Hernan Garcia, Jose Martin Hernandez-Ayon, Xinping Hu, Alex Kozyr, Chris Langdon, Kitack Lee, Joseph Salisbury, Zhaohui Aleck Wang and Liang Xue (2021) "Best Practice Data Standards for Discrete Chemical Oceanographic Observations." Frontiers in Marine Science, in review.*

[O6] line 66: "in-water pressure"? Do the authors mean the "hydrostatic pressure"?

*Yes, updated.*

[O7] line 83: The authors should clarify what is meant but "total salt contents". It would be useful to see the full list in parentheses.

*Added list in parentheses "(ammonia, borate, calcium, fluoride, phosphate, silicate, sulfate, and sulfide)"*

[O8] line 131-132: The phrase "because these are all directly proportional to each other" may mislead readers because the relationships between the terms mentioned depend on temperature, salinity, and atmospheric pressure. Thus it would be useful to add at the end of the sentence, "at a given temperature, salinity, and atmospheric pressure".

*Updated following reviewer's suggestion.*

[O9] line 159, 162: Use of "every" may be misunderstood by readers. PyCO2SYS does not account for alkalinity from organic acids except for advanced users that have the ability to specify the appropriate parameters for their regional conditions. And even they will not be able to precisely add in all the missing components.

*We updated the first occurrence to 'every component of alkalinity in the main chemical speciation equation' and the second to 'every modelled component'.*

**3.1.3 pH SCALE CONVERSIONS**

[O10] line 197: "negligible"? Please provide the readers with approximate quantitative estimates of what the differences in pH from this error actually are at typical seawater pH when the user selects the free and seawater pH scales.

*Changed parenthetical phase in this line to: "(because $[HSO_4^-]$ and $[HF]$ are each on the order of $10^{-10}$ umol/kg, relative to AT on the order of 2000 umol/kg)"*

**3.3.2 GAS CONSTANT**

[O11] line 276: Please be more quantitative.  Some might take the qualitative "minor" to be on the order of a percent or more, but the actual effect on calculated variables from changing the 4th decimal point of R must be very much smaller.

*Added "less than $10^{-4}$%" for the pCO2, fCO2, and xCO2 conversions and "less than $10^{-3}$% at 5000 dbar" for the pressure corrections for equilibrium constants. These values were determined by running CO2SYS-MATLAB with the old R value and new R value.*

**3.3.4 BUFFER FACTORS**

[O12] lines 296-298: The typographical errors in the equations by Egleston et al. were corrected well before 2018. They were first corrected by Orr (2011), who provided a function in seacarb \textit{buffesm}, released also in 2011, that calculated the Egleston et al. buffer factors but with the corrected equations. Alvarez et al. (2014) later provided equations the same corrections. Those earlier fixes were mentioned in section 2.2 of Orr et al (2018).

*Thank you, we have added these earlier references.*

[O13] lines 298-299: The authors are right to mention that the Egleston et al. equations "do not include the effect of species beyond the carbonate, borate, and water contributions to total alkalinity." However, these equations were extended by Orr et al. (2018) to also include contributions to total alkalinity from phosphoric and silicic acid systems. See their Appendix B. They also updated their \textit{buffesm} routine, available in both seacarb and mocsy, to include these extensions.

*We added a note that Orr et al. (2018) have extended these equations to include phosphate/silicate effects.*

[O14] lines 304-306: Another reviewer, Guy Munhoven, seems to suggest adding an option for computing the analytical solution for the Revelle factor. This solution, with the corrected equations from Egleston et al. (2010) and the Orr et al (2018) extensions to include the phosphoric and silicic acid

contributions to total alkalinity is available in the \textit{buffesm} function of both seacarb and mocsy.

*This refers to comment [M15]. Thank you for the pointer and please see our response there.*

**3.4 NO-SOLVE MODES**

[O15] line 318: It would be useful to mention what assumption is made in PyCO2SYS about atmospheric pressure when converting to and from xCO2 (Patm = 1 atm). This may be a large correction, particularly in the very high latitudes, and CO2SYS does not account for it.

*Thank you for pointing this out. Atmospheric pressure can now be provided as an input to PyCO2SYS for these conversions. See also [O30].*

**3.6 UNCERTAINTY PROPAGATION**

[O16] line 332: "This feature" seems ambiguous. Do the authors mean "Propagating the uncertainty" or "Calculating the derivative"?

*Updated to "Uncertainty propagation"*

[O17] line 335: what is the difference between "totally independent" and just "independent"? Or perhaps one could just say "differs".

*"Totally independent" is just the writer being overdramatic. Updated to "differs".*

[O18] line 336: I did not realize that "automatic differentiation, in general, was so computationally inefficient. Over what parts of PyCO2SYS is automatic differentiation inefficient? It would be very nice if the authors could provide quantitative statements regarding this statement.

*See response to reviewer comment [M17]. AD itself is not necessarily so inefficient, but the most convenient implementation currently suitable for PyCO2SYS is a bit slow over long and complicated functions with lots of control flow elements. Improving this is a goal for the future as*

*mentioned in Section 5.5 but not essential for PyCO2SYS to be a useful tool for the community.*

[O19] Equation (23): I am a little surprised by this scaling, which would seem to be applied almost always since the median of "a" would almost never be exactly zero. What about very small numbers, where the median was close to but not quite zero? This would make the step size much too small. The choice of Δa seems arbitrary. Have the authors tried to seek out optimal values by comparing their partial derivatives calculated by finite differences with those that they could calculate from the analytical solutions based on the buffer factors from Egleston et al.?

This scaling by median(a) also means that the uncertainties for a particular line of input data will depend on the values of the other lines in the data set. This makes calculations less reproducible.

*Good point. We agree. We have implemented a fixed Δa value instead. See updated discussion in this section and the new Appendix.*

[O20] line 355: Equation 24 is incorrect. It is missing the partial derivatives, something which might be suspected because the units of the left-side term with differs with each right-hand side term. See equation (1) in Orr et al. (2018).

*Thanks, this is a typo in the manuscript, which we have now corrected. It was implemented correctly in the code.*

[O21] line 358: The authors state, "The user can therefore build". This phrase is not clear. Does PyCO2SYS actually offer an easy way for the user to compute a set of covarying argument uncertainties? This is a tricky business because correlations between the *uncertainties* of various input variables are hard to ascertain. Does PyCO2SYS offer something original in this regard, beyond what is now available in other packages?

*PyCO2SYS can provide every derivative one might need. The task of assembling these into a Jacobian matrix is left to the user and this is what the unclear phrase refers to. The reviewer rightly notes that it's difficult to determine covariances in uncertainties between different*

*arguments and few researchers actually (are able to(?)) do this. Therefore it has not been a priority to create a generalised, user-friendly function for this in PyCO2SYS. We may add something in the future but we don't think it's necessary for this initial publication. The key point is that for now, someone who does know what they are doing (in terms of both maths and Python) can easily get the necessary partial derivatives from PyCO2SYS.*

**4.1.2 BUFFER FACTORS**

[O22] lines 380-390:  Just after this subsection it would be useful to insert another subsection where the partial derivatives calculated by finite differences would be compared to those that could be calculated from the buffer factors of Egleston et al. (either the automatic derivatives or the analytical solutions). See Orr et al. (2018, equations B9 and B13), which details the conversion between buffer factors and partial derivatives.

*This is a nice suggestion that we will keep in mind for future developments to the software and its test suite, but we don't see that this adds a new dimension to what is already being tested, but rather replicates information from existing tests within PyCO2SYS, so we don't find it necessary for the current version and this publication.*

**4.1.3 UNCERTAINTY PROPAGATION SIMULATIONS**

[O23] The agreement of 3% is not bad, but I think it could be much better if the authors would use more iterations. Based on previous such comparison (Orr et al., 2018, Fig. 3), I would suggest that the authors try to repeat this comparison but use $n=10^6$ instead of $n=10^4$.

*Yes, increasing the number of iterations does improve the agreement, confirming that this is the reason why we need such a large cutoff value (3%) rather than it being due to some problem in the code. But running 10^4 iterations already takes a few minutes and this scales linearly, so 10^6 iterations would make the test suite impractical to run on a regular basis, therefore we have left this as it is.*

[O24] line 441: I struggled to understand "from one of these, [H+], and K1 and K2 equilibrium constants using the equations in Appendix C".  Please try to clarify.  I would suggest (if it is what is actually meant by the authors) to change "these" to "these three variables". Also, I would recommend to be more specific by referring to the 3 relevant subsections of Appendix C, rather than the whole Appendix, which is quite long.

***The reviewer's understanding is correct. We have updated the text following these suggestions.***

[O25] lines 451-453:  Differences of 20% between this study's estimates and those from Orr et al. (2018) for partial derivatives of [H+] do not seem negligible, unlike what is suggested by the authors. I do not believe that these differences are due to the "pH scale conversion simplification (Sect. 3.1.3)" as suggested by the authors. Indeed, those derivatives from Orr et al. (2018) were calculated for the total [H+] scale, for which the authors say this issue presents no problem. The authors also suggest that the 20% difference could come from "other differences in pH-solving from AT and TC (Sect. 3.1)", but I still don't understand how.  In Orr et al. (2018), we got the same results for the partial derivatives in multiple packages (seacarb, CO2SYS, mocsy) as well as with different approaches in the same package, including finite differences, automatic derivation and analytical (symbolic) solutions. Furthermore, because [H+] and pCO2 are nearly linearly related, it does not make sense to me that the authors find a 20% difference for partial derivatives of [H+] but not for partial derivatives of pCO2. Did the authors use the free pH scale for the calculations of the partial derivatives of [H+]?

***Oops. The 20% difference is because we did not realise the [H+] reported in the Orr paper was on the Total scale, so we were using Free scale as suggested by the reviewer. We converted to Total scale and now agreement is the same as for all other variables. Thank you for pointing this out!***

**5.3 PRESSURE CORRECTIONS FOR pCO2**

[O26] lines 529-531: The authors might mention that these pressure corrections have already been implemented in two other packages, seacarb

and mocsy, see Orr et al. (2015, Tables 9 and 10), as well as Orr and Epitalon (2015).

*We have added this mention.*

**APPENDIX B:**

[O27] line 627: The authors state, "Undissociated H2CO3 is considered negligible and thus not modelled (Zeebe and Wolf-Gladrow, 2001)." Actually, H2CO3 is modelled, but not separately.  The undissociated carbonic acid [H2CO3] is indeed a small fraction of the total, but in the CO2 system equations, it is actually taken into account together with the dissolved CO2 (molecular or aqueous CO2). Please refer to the best practices guides (Dickson et al., 2007; Dickson, 2010).  For instance, in Dickson et al. (2007) Chapter 2 states, "It is usual to combine the concentrations of CO2(aq) and H2CO3(aq) and to express this sum as the concentration of a hypothetical species, CO2(aq)." Thus in all equations where the authors have used CO2(aq), that should be replaced by CO2*(aq) or simply CO2*.

*Updated this sentence to note that undissociated H2CO3 is included within the CO2(aq) term.*

**APPENDIX C:**

[O28] line 689: "in some cases, simpler alternatives are used instead".  This is vague. Can the authors be more specific as to exactly where simplifications are used (and where they come from).  These details could be given when the simplifications themselves are presented. It might also be appropriate to mention the best-practice guides from Dickson et al. (2007) and Dickson (2010) in addition to Zeebe and Wolf-Gladrow (2001).

*We removed this vague sentence as it was not really necessary, and added the reference to the best-practice guide. For the reviewer's interest we were referring to ZWG's cases (2), (3) and (10) which are where the known pair are two from [CO2(aq)], [HCO3] and [CO3]. In each case ZWG generate a cubic or quartic equation of [H+] to be solved. However much simpler expressions to get a third core parameter can be*

*generated by simply combing the definitions of K1\* and K2\* and rearranging (e.g. [CO3] = [HCO3]^2 K1 / ([CO2(aq)] K2)).*

**C1.2 KNOWN pCO2, xCO2 or [CO2(aq)]**

[O29] line 704: I have not seen the name "vapour pressure factor" before. Does that even have to be named? If so, why not a simpler term, and one that is more relatable, such as "humidity correction" or "wet-air correction".

*The name came from the terminology used in functions and code comments in the original CO2SYS code. We have switched to 'humidity correction' in the manuscript (here and in Table 3).*

[O30] line 706: Can the authors please clarify what is meant by "(assumed to be 1 atm)"? Do users have a choice for P (an input argument) or is the value of P=1 atm hard wired into the code. As mentioned by Orr et al. (2017), "the average surface atmospheric pressure between 60 and 30S is 3% lower than the global mean, thus reducing surface-ocean pCO2 by 10 uatm". This seems like a big correction, and it would be nice for users to know if PyCO2SYS provides for it.

*Thank you for pointing this out. P = 1 atm was hard-wired into the v1.7 code. However, we have now updated the PyCO2SYS code such that a different value for barometric pressure can be provided instead, while 1 atm is still used by default. This is described in new Section 3.3.5. See also [O15].*

**APPENDIX F:**

[O31] Please be more quantitative? What does "mostly negligible" mean? For example, to what decimal place is the Revelle factor typically affected by these errors?

*First decimal place. We have updated the section to state this.*

**APPENDIX G: PROPAGATION OF CO-VARYING UNCERTAINTIES**

[O32] lines 825-837: This appendix too greatly resembles Appendix A from Orr et al. (2018). It has the same equations and similar text. It is not original and should be deleted. The appropriate citation could be given in the text. Overall, the authors do not do justice to the complicated subject of propagating uncertainties when uncertainties of the different input parameters are correlated, and just adding this appendix is not a good solution. Referring to previous work and mentioning what new features or facility that PyCO2SYS offers would seem a better option.

***We do not claim these equations to be original; as noted in the final paragraph of the Introduction, 'Equations that were... taken from the literature are reported in appendices'. We note also that the equations and explanation we provided were written without reference to the Orr et al. appendix, which is equally not original. We have replaced our appendix with a reference to the GUM. We agree that we do not 'do justice' to the topic of propagating correlated uncertainties, indeed we have made no attempt to do so, as (1) PyCO2SYS does not include this feature and (2) uncertainty propagation is not the main focus of this manuscript (see also our response to [O21]).***

**GLOBAL CHANGE**

[O33] For consistency with the best-practice guides and with the $A_T$, I would recommend to change $T_C$ symbol to $C_T$.

***There is (unfortunately) no universal standard for these symbols in the research field. Quite some thought went into the symbols in this manuscript with reference to the IUPAC Green Book, and we believe that the current approach is the most consistent both with that reference and also internally in this manuscript.***

***The main problem with C_T is that an equivalent approach cannot be used for the other elements. P_T is possible but Si_T is not a valid symbol (multiple italicised main characters; italicised element symbol)***

*following the Green Book. We also must distinguish between totals of SO_4 and H_2S, so cannot simply use S_T. Furthermore, following the Green Book, chemical formulae should not be italicised (e.g. C representing carbon). It would be incongruous to use C_T for DIC but T_X for all the others.*

*Our approach is thus. The main symbol refers to the overall concept. So A means alkalinity, T means total substance content. The subscript shows "of what" in both cases. For example, A_C is the alkalinity associated with all inorganic C species. T_C is the total content of all inorganic C species. A_T is a special case being the total alkalinity across all equilibrium systems.*

*See e.g. Appendix B in which each subsection has an equation for each of T_X and A_X for the relevant system.*

**PUNCTUATION/LANGUAGE (minor and few)**

[O34] * The colon is used incorrectly in lines 29, 90, 146, 215, 223, 233, 248, 769, 774, 792, 800, 817.  This misuse has been referred to as the most common colon erorr ever. In all these cases, the colon should just be deleted.

*We thank the reviewer for the tip and have adjusted these sentences as suggested.*

[O35] * line 167: "too" could be deleted.

*Deleted "too".*